# Bridging Human and LLM Judgments: Understanding and Narrowing the Gap

**Felipe Maia Polo**[1][*], **Xinhe Wang**[1][*], **Mikhail Yurochkin**[2]
**Gongjun Xu**[1], **Moulinath Banerjee**[1], **Yuekai Sun**[1]
[1]Department of Statistics, University of Michigan
[2]Institute of Foundation Models, MBZUAI

## Abstract

Large language models are increasingly used as judges (LLM-as-a-judge) to evaluate model outputs at scale, but their assessments often diverge systematically from human judgments. We present `Bridge`[1], a unified statistical framework that explicitly bridges human and LLM evaluations under both absolute scoring and pairwise comparison paradigms. `Bridge` posits a latent human preference score for each prompt-response pair and models LLM deviations as linear transformations of covariates that capture sources of discrepancies. This offers a simple and principled framework for refining LLM ratings and characterizing systematic discrepancies between humans and LLMs. We provide an efficient fitting algorithm with asymptotic guarantees for statistical inference. Using six LLM judges and two benchmarks (BigGen Bench and Chatbot Arena), `Bridge` achieves higher agreement with human ratings (accuracy, calibration, and KL divergence) and exposes systematic human-LLM gaps.

## 1 Introduction

Accurate and reliable evaluation is fundamental to the advancement and deployment of artificial intelligence (AI) systems, such as Large Language Models (LLMs). Traditional expert-based or automatic methods (*e.g.*, ROUGE [27] or BLEU [33]) for open-ended generated text often struggle with either scalability or poor quality. Recently, LLMs have shown significant promise in addressing these challenges through the emergent "LLM-as-a-Judge" (LLMJ) paradigm [12, 22, 23]. Leveraging their extensive knowledge, flexible reasoning, instruction following, and natural language understanding, LLMs can effectively evaluate complex tasks by scoring, ranking, or selecting among diverse outputs. However, ensuring the trustworthiness, robustness, and human-alignment of LLM-based evaluation systems remains a critical challenge that necessitates careful attention.

A crucial step towards better judges involves deepening our understanding of LLMJ systems and recognizing their inherent strengths and limitations. Such knowledge allows practitioners to better quantify associated risks and steer the development of more robust evaluation methods. Recent research has extensively explored factors contributing to inaccuracies in LLM judgments and proposed ways to mitigate them. For example, various biases have been well-documented, including preferences for lengthier responses, overly generous scoring tendencies, or biases influenced simply by the presentation order during pairwise comparisons [8, 41, 48, 39]. In this work, we introduce `Bridge`, a statistical framework that explicitly connects LLM ratings to human judgments, providing deeper insight into the sources of human-LLM discrepancies and enabling more reliable alignment between

---

[*]These authors contributed equally to this work. Corresponding authors: felipemaiapolo@gmail.com; xinhe.wang07@gmail.com
[1]Please check our GitHub repository: https://github.com/felipemaiapolo/bridge

39th Conference on Neural Information Processing Systems (NeurIPS 2025).

the two. Our framework is LLM-agnostic, does not require access to model weights, and can be applied on top of any API.

`Bridge` combines a statistical model with a specialized fitting algorithm via the proposed *logit trick*, and we demonstrate the asymptotic normality of the resulting estimators. Our model assumes that both human annotators and LLM judges score each prompt-response pair according to a shared latent preference signal, while systematic deviations in LLM scores are captured by a linear transformation of covariates that encode potential human-LLM divergence sources (*e.g.*, response length, text sentiment, writer's creativity). This formulation enables simultaneous, rigorous, interpretable estimation and testing of *multiple discrepancies between human and LLM judgments*; something which current approaches cannot accomplish. In addition, it enables lightweight post-hoc corrections: with only a small set of human labels, we can recalibrate LLM scores for improved probabilistic alignment with human assessments.

In summary, our contributions are:

1. Proposing `Bridge`, a statistical framework connecting human and LLM judgments, which combines a statistical model with a specialized fitting algorithm via the proposed *logit trick*. Our approach (i) allows practitioners to better understand what makes humans and LLMJ different, (ii) enables better probabilistic alignment with human judgments, and (iii) is LLM-agnostic, making it applicable on top of any API.

2. Deriving the asymptotic distribution of our parameter estimators. These asymptotic distributions allow us to construct confidence intervals for our parameters and formally test for different types of human-LLM gaps. Moreover, it allows us to construct confidence intervals for predictive quantities such as the probability of humans making a certain judgment.

3. Validating our framework using six different LLM judges and queries from BigGen Bench and Chatbot Arena. We show that we can better align LLMs to human annotators using a few labeled data points and interpret the systematic differences between the two types of judges.

## 1.1 Related work

Improving the alignment between LLM-based judges and human annotators is often an important factor for their reliable use at scale. Prior work has pursued several complementary strategies: supervised fine-tuning on human-labelled data [14]; supplying in-context examples to the judge [48]; post-hoc smoothing of raw scores [28, 21]; decomposing complex evaluation tasks into simpler, verifiable criteria [38, 44]; harnessing reference answers or detailed rubrics [17, 47], or chain-of-thought (CoT) strategies to make use of the model's reasoning abilities [17, 25].

A parallel line of research first diagnoses systematic biases in LLMJ and then proposes corrective measures. For instance, Dubois et al. [8] reveals a strong preference for longer responses and introduces the length-controlled AlpacaEval [26] benchmark. Park et al. [34] catalogue six distinct bias types, construct counter-biased datasets, and fine-tune judges on these counter-examples. Additional studies on LLMJ biasing factors document, for example, position bias [46, 48, 39, 43], leniency bias [41], and sentiment or authority biases [46]. Few works have also uncovered biases in human ratings themselves [5, 24]. On a related but different direction, Buyl et al. [4] studies how discretion is exercised by humans and LLMs when making choices based on key guiding principles.

Despite recent advances, no prior work has systematically examined divergences between human and LLM judgments in a comprehensive way. These discrepancies may carry negative connotations (*e.g.*, LLM biases), be value-neutral, or even positive (*e.g.*, human biases). Yet such a comparison is essential: it clarifies the strengths and limitations of LLM judges and underscores that corrections should be made *relative* to human preferences, not in absolute terms, if human alignment is the goal. Otherwise, eliminating a feature valued by both humans and LLMs could inadvertently *widen* the misalignment. To fill this gap, our work offers a principled way to analyze the systematic differences between human and LLM judges.

## 2 Problem setup

We consider two evaluation scenarios: absolute and relative ratings. In the absolute case, an entity under evaluation (*e.g.*, an LLM or a human) is presented with an input prompt $I$ and produces a

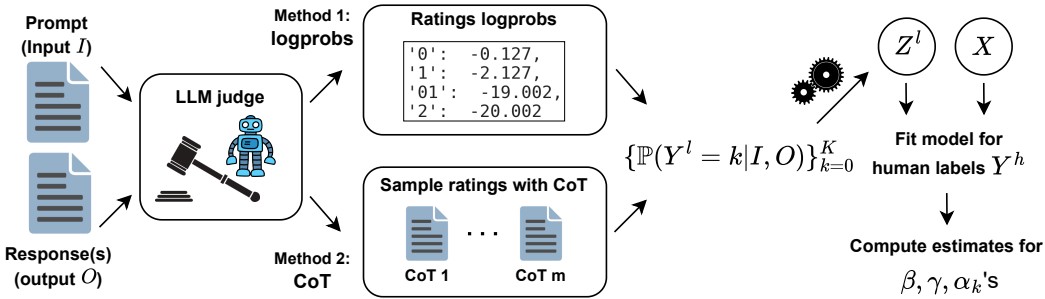

Figure 1: The *logit trick* for model fitting. This procedure allows us to fit the statistical model without observing human latent scores $Z^h$. First, the LLM judge rates a pair (prompt, response(s)). Second, we compute/estimate the probability of each score $k$. Third, we process the probabilities, obtaining the LLM scores $Z^l \in \mathbb{R}$. Finally, we fit an ordinal logistic regression model for human ratings $Y^h$ given $Z^l$ and covariates $X$ to explain the gap between human and LLM scores.

text output $O$[2]. Given the pair $(I, O)$, both a human evaluator and an LLM judge assign scalar ratings $Y^h, Y^l \in \mathbb{R}$, where the numerical values reflect absolute assessments, such as whether the response is satisfactory. In the relative case, two entities generate responses $O_A$ and $O_B$ to the same prompt $I$, and evaluators provide comparative judgments $Y^h$ and $Y^l$ that encode one of the options in $\{O_A \text{ wins, tie}, O_B \text{ wins}\}$. We denote the output by $O = (O_A, O_B)$ in this setup.

Our goal is to develop a statistical framework, `Bridge`, that bridges human and LLM scores, $Y^h$ and $Y^l$, under two key assumptions: (i) both are influenced by a shared latent score $Z^h = f(I, O) \in \mathbb{R}$ representing human preferences, and (ii) LLM judgments may be systematically different from human judgments due to additional features encoded in a covariate $X = g(I, O) \in \mathbb{R}^d$. The map $g$ is assumed to be known, and it can, for example, capture properties of the output $O$, such as formatting (*e.g.*, markdown usage), structural attributes (*e.g.*, number of paragraphs), stylistic elements (*e.g.*, sentiment), or text-quality attributes such as creativity and factuality. `Bridge` enables both the refinement of LLM judges and the analysis of discrepancies between human and LLM evaluations.

## 3 The `Bridge` framework

### 3.1 Statistical model and fitting

**The model.** We consider discrete judgments $Y^h, Y^l \in \{0, \cdots, K\}$, where ratings reflect a clear ordinal structure. For absolute scoring, these judgments reflect ordered levels of satisfaction or agreement, so that a rating $Y^h = k + 1$ is preferred over $Y^h = k$. For relative scoring, each level $k$ captures the strength of preference between two responses. For instance, when $K = 2$, a judgment of $Y^h = 0$ may indicate that $O_A$ is preferred to $O_B$, $Y^h = 1$ denotes no preference (a tie), and $Y^h = 2$ indicates preference for $O_B$, giving rise to a Bradley-Terry-type model [3, 37].

To model $Y^h$ and $Y^l$, we use the ordinal logistic regression (ordered logit) formulation [45]. For human judgments $Y^h$, we assume the model $\mathbb{P}(Y^h = k \mid I, O) = p_k(\alpha_1, \ldots, \alpha_K, Z^h)$, where

$$p_k(\alpha_1, \ldots, \alpha_K, Z^h) \triangleq \begin{cases} \sigma(\alpha_1 - Z^h), & \text{if } k = 0, \\ 1 - \sigma(\alpha_K - Z^h), & \text{if } k = K, \\ \sigma(\alpha_{k+1} - Z^h) - \sigma(\alpha_k - Z^h), & \text{otherwise,} \end{cases}$$

$\sigma$ is the standard logistic function (*i.e.*, sigmoid function), $\alpha_k < \alpha_{k+1}$ are ordered real cutoffs, and $Z^h = f(I, O)$ is a latent factor representing human preferences. The corresponding model for LLM judgments $Y^l$ replaces cutoffs $\alpha_k$ with $\eta_k$ and $Z^h$ with $Z^l$, assuming $Z^l \triangleq \beta Z^h + \gamma^\top X$, where $X = g(I, O) \in \mathbb{R}^d$ captures features associated with deviations between LLM and human evaluations. The function $g$ may represent either engineered features (*e.g.*, text length or sentiment) or learned representations (*e.g.*, neural embeddings of $(I, O)$). This formulation allows us to model systematic differences and refine LLM outputs accordingly.

---

[2]While $O$ may take other forms in principle, we restrict our attention to textual outputs.

**Model fitting via the *logit trick*.** Given a pair $(I, O)$, if the score $Z^l$, parameters $\beta, \gamma$, and cutoffs $\{\alpha_k\}_{k=1}^K, \{\eta_k\}_{k=1}^K$ were known, we could directly analyze the gap between judges and adjust LLM predictions to align with human preferences. However, none of these quantities are observed in principle. Moreover, the human latent scores $Z^h$ are unobserved, which makes parameter estimation appear intractable, even assuming access to $Z^l$. Nevertheless, when human labels $\{Y_i^h\}_{i=1}^n$ are available for a set of input-output pairs $\{(I_i, O_i)\}_{i=1}^n$, we can leverage a technique we call the ***logit trick*** to circumvent this issue. For now, we assume $\mathbb{P}(Y_i^l = k \mid I_i, O_i)$ for all $k \in \{0, \dots, K\}$ can be computed exactly (or estimated with high precision). Details on how this step is implemented are provided later in this section. The model fitting algorithm via the *logit trick* is detailed in the following box:

---

**Model fitting via the *logit trick***

1. For each example $(I_i, O_i)$, compute $\mathbb{P}(Y_i^l = k \mid I_i, O_i)$ for all $k \in \{0, \dots, K\}$.

2. Compute $Z_i^l$ and cutoffs $\{\eta_k\}_{k=1}^K$ by solving:

$$\left(\{\eta_k\}_{k=1}^K, \{Z_i^l\}_{i=1}^n\right) = \underset{\substack{\bar{\eta}_1 < \cdots < \bar{\eta}_K \in \mathbb{R}, \\ z_1, \dots, z_n \in \mathbb{R}}}{\arg\min} \sum_{i=1}^n \sum_{k=0}^K \left| p_k(\bar{\eta}_1, \dots, \bar{\eta}_K, z_i) - \mathbb{P}(Y_i^l = k \mid I_i, O_i) \right|.$$

3. Fit the ordinal logistic model to the human labels $\{Y_i^h\}_{i=1}^n$ using maximum likelihood, with $Z_i^h = (1/\beta)Z_i^l - (1/\beta)\gamma^\top X_i$ as inputs, *i.e.*,

$$\left(\{\hat{\alpha}_k\}_{k=1}^K, \hat{\beta}, \hat{\gamma}\right) = \underset{\substack{\alpha_1 < \cdots < \alpha_K, \\ \beta \in \mathbb{R}, \gamma \in \mathbb{R}^d}}{\arg\max} \sum_{i=1}^n \sum_{k=0}^K \mathbf{1}\{Y_i^h = k\} \log p_k\left(\alpha_1, \dots, \alpha_K, (1/\beta)Z_i^l - (1/\beta)\gamma^\top X_i\right).$$

---

When the model for $Y^l$ is correctly specified, the optimization in step 2 recovers the true values of $\{\eta_k\}_{k=1}^K$ and $\{Z_i^l\}_{i=1}^n$ up to an additive constant. To ensure identifiability, we fix $\eta_1 = 0$. As a simpler alternative, one may define $Z_i^l = -\sigma^{-1}(\mathbb{P}(Y_i^l = 0 \mid I_i, O_i))$, but this approach uses only one probability and ignores the full distribution[3] of $Y^l$, which can be suboptimal under model misspecification. At test time, new values of $Z^l$ can be derived using the estimated thresholds $\{\eta_k\}_{k=1}^K$ and solving an optimization like in step 2. A better option, if the application permits, is computing test points $Z^l$'s jointly with those from training data. After the model is fitted, we can define the predicted human latent scores as $\hat{Z}^h \triangleq (1/\hat{\beta})Z^l - (1/\hat{\beta})\hat{\gamma}^\top X$.

We consider two strategies for computing $\mathbb{P}(Y^l = k \mid I, O)$. The first is based on log probabilities and return exact values: we identify the tokens associated with each possible outcome $k$, compute their probabilities from the LLM output distribution, and sum them. This method is computationally efficient and provides exact probabilities, but it requires prompting the LLM to output the final score without intermediate reasoning. When reasoning steps are used, these probabilities may become biased[4]. As an alternative, we employ a chain-of-thought (CoT) prompting strategy, in which the LLM produces reasoning followed by a rating. In this case, we sample $m$ outputs from the LLM and estimate $\mathbb{P}(Y^l = k \mid I, O)$ via empirical frequencies. In this case, $\mathbb{P}(Y^l = k \mid I, O)$ is not computed exactly. While this approach is more computationally intensive, it typically yields higher-quality judgements by leveraging the model's reasoning capabilities. In both strategies, we regularize the output probabilities by adding a small constant (e.g., 0.01) to each $\mathbb{P}(Y^l = k \mid I, O)$ before renormalizing to ensure they sum to one and avoid degenerate distributions.

Figure 1 provides a visual overview of the entire procedure, and Appendix A contains the prompt templates used to collect LLM judgements in both the log probabilities and CoT cases.

---

[3] When $Y^h, Y^l \in \{0, 1\}$, however, these two approaches are equivalent. This observation also clarifies why we call it the *logit trick*: under the binary judgements, $Z^l$ is precisely the logit of $\mathbb{P}(Y^l = 1 \mid I, O)$.

[4] Once the LLM generates reasoning steps $R$ (a sequence of tokens), its subsequent score is conditioned on $R$, biasing the probabilities toward outcomes that are more compatible with that specific reasoning. To avoid this bias, we marginalize over all possible reasoning paths, *i.e.*, $\mathbb{P}(Y^l = k \mid I, O) = \sum_r \mathbb{P}(Y^l = k \mid I, O, R = r)\mathbb{P}(R = r \mid I, O)$, and estimate this sum via Monte Carlo sampling rather than relying on a single conditional probability $\mathbb{P}(Y^l = k \mid I, O, R = r)$.

**Model extensions.** We work under the setup in which $Y^h$ and $Y^l$ are discrete and obey a notion of ordering because it covers the majority of practical use cases, including binary ratings (*i.e.*, $Y^h, Y^l \in \{0, 1\}$). However, it does not account for continuous or unordered categorical ratings. We develop extensions of our model to deal with those cases, and, due to space constraints, we include them in Appendix E.

## 3.2 Non-exhaustive set of applications

**Better alignment and calibration.** A key application of our framework arises when practitioners seek to improve the quality of LLM-generated judgments, especially in low-resource settings where human-labeled data is limited due to the high cost of annotation. In such scenarios, fine-tuning LLM judges is often impractical or even impossible when inference APIs are used. We propose using our model to enhance both the alignment and probabilistic calibration of LLM judgments. Alignment between LLM and human judgments is crucial for enabling high-quality evaluations at reduced cost. Calibration is equally important, as well-calibrated models yield more interpretable outputs, facilitating uncertainty quantification. Moreover, well-calibrated scores do not suffer from position bias (relative to humans) by definition, for example.

We quantify alignment by measuring the discrepancy between LLM-inferred rating probabilities and the target distribution $\mathbb{P}(Y^h = k \mid I, O)$, using human labels and cross-entropy loss. Specifically, we expect that the model-implied probabilities $p_k(\hat{\alpha}_1, \ldots, \hat{\alpha}_K, \hat{Z}^h)$ more closely approximate $\mathbb{P}(Y^h = k \mid I, O)$ than the raw LLM outputs $\mathbb{P}(Y^l = k \mid I, O)$ in terms of the Kullback-Leibler (KL) divergence [20]. Additionally, we also check alignment in terms of accuracy. For calibration, we adopt the class-wise notion [35], which requires

$$\mathbb{P}(Y^h = k \mid p_k(\hat{\alpha}_1, \ldots, \hat{\alpha}_K, \hat{Z}^h) = p) \approx p \quad \text{for all } p \in [0, 1], \ k \in \{0, \ldots, K\}.$$

A property like this is unlikely to hold for raw LLM predictions, but becomes more plausible when LLM scores are corrected using our model, assuming it is reasonably well-specified. Analogous calibration methods are common in classification tasks, such as Platt scaling [36], where a logistic regression is applied to uncalibrated classifier scores to improve their probabilistic interpretation.

As we demonstrate in our experiments, even under the simplifying assumption $\gamma = 0$, *i.e.*, without using any covariates, our model yields improved LLM judgment predictions at test time, highlighting its practical utility in resource-constrained settings.

**Human-LLM discrepancies quantification and formal testing.** Another important application is the detection and quantification of human-LLM judgement divergences. To that end, we assume $X$ contains possible sources of differences between LLM and human judgements, and we want to better understand which differences are relevant by analysing $\gamma$. As detailed in Section 3.3, the estimator $\hat{\gamma}_j$ for the $j$-th entry of $\gamma$ is asymptotically normal, *i.e.*, $(n/\hat{V}_{j+1,j+1})^{1/2}(\hat{\gamma}_j - \gamma_j)$ converges in distribution to $\mathcal{N}(0, 1)$ as $n \to \infty$, where $\hat{V}_{j+1,j+1}$ is a variance estimate derived from the data. This result enables formal hypothesis testing[5], *e.g.*, testing $H_0 : \gamma_j = 0$ versus $H_1 : \gamma_j \neq 0$, using the p-value

$$\text{p-value} = 2\Phi\left(-\sqrt{n/\hat{V}_{j+1,j+1}} \, |\hat{\gamma}_j|\right),$$

where $\Phi$ denotes the distribution function of the standard normal distribution. Additionally, if the practitioner wants to control the false discovery rate (FDR) of human-LLM divergences, multiple hypothesis testing can be carried out in conjunction with the Benjamini-Yekutieli procedure [2]. Confidence intervals can also be constructed, as discussed in Section 3.3.

## 3.3 Asymptotic distributions of our estimators

First, we analyze the properties of estimators for the cutoffs $\{\eta_k\}_{k=1}^K$ and latent scores $\{Z_i^l\}_{i=1}^n$ under the CoT prompting strategy for estimating $p_{ik} = \mathbb{P}(Y_i^l = k \mid I_i, O_i)$. Fix $n$ evaluation samples $\{(I_i, O_i)\}_{i=1}^n$. For each $i$, draw $m_n$ i.i.d. CoT judgements $\{Y_{i,m}^l\}_{m=1}^{m_n}$ and estimate $p_{ik}$ with $\hat{p}_{ik,m_n} = \sum_{m=1}^{m_n} \mathbf{1}\{Y_{i,m}^l = k\}/m_n$. Denote $\eta = (\eta_1, \ldots, \eta_K)$, and define the empirical and population losses as

$$Q_{n,m_n}(\eta, z_{1:n}) = \sum_{i=1}^n \sum_{k=0}^K |p_k(\eta, z_i) - \hat{p}_{ik,m_n}|, \quad Q_n(\eta, z_{1:n}) = \sum_{i=1}^n \sum_{k=0}^K |p_k(\eta, z_i) - p_{ik}|.$$

---

[5]Here, rejecting $H_0$ means feature $j$ systematically shifts the LLM's judgments relative to humans.

Constrain $\eta, Z_{1:n}^l$ to $\Theta_\eta = \{\eta \in \mathbb{R}^K : 0 = \eta_1 < \cdots < \eta_K\}$ and $\mathcal{Z} = [-M, M]^n$ for some large $M$.

**Proposition 3.1** (Consistency of CoT estimates $\hat{\eta}_k$ and $\hat{Z}_i^l$). *Under Conditions C.1 and C.2 (stated in Appendix C.1), the estimator $(\hat{\eta}, \hat{Z}_{1:n}^l) \in \arg\min_{(\eta, z_{1:n}) \in \Theta_\eta \times \mathcal{Z}} Q_{n,m_n}(\eta, z_{1:n})$ satisfies $\sqrt{m_n}[(\hat{\eta}, \hat{Z}_{1:n}^l) - (\eta^*, Z_{1:n}^{l,*})]$ converges in distribution to a mean-zero distribution with covariance $\Sigma$ (defined in Appendix C.1) as $m_n \to \infty$.*

When the log probabilities are used to extract $p_{ik}$ exactly from the LLM output, the population loss $Q_n$ is known and $\hat{Z}_i^l = Z_i^l$ for all $i$. Next, we derive the asymptotic distribution of $(\hat{\beta}, \hat{\gamma})$ as $n$ tends to infinity, under either log probability or CoT-based estimation of the latent scores. Write the ordered logit model as $\mathbb{P}(Y_i^h = k \mid I_i, O_i) \triangleq l_k(\theta; Z_i^l, X_i) = p_k(\alpha_1, \ldots, \alpha_K, (1/\beta)Z_i^l - (1/\beta)\gamma^T X_i)$. Define the Fisher information matrix $\mathcal{I}(\theta^*) = -\mathbb{E}\big[\nabla_\theta^2 \log l_{Y_i^h}(\theta^*; Z_i^{l,*}, X_i)\big]$.

**Theorem 3.2** (Asymptotic normality of $(\hat{\beta}, \hat{\gamma})$). *Under Conditions C.3–C.5 (stated in Appendix C.1), let the MLE $\hat{\theta}_n = (\hat{\alpha}_1, \ldots, \hat{\alpha}_K, \hat{\beta}, \hat{\gamma})$ maximize the log-likelihood*

$$\ell_n(\theta; \hat{Z}^l, X) = \sum_{i=1}^{n} \sum_{k=0}^{K} \mathbf{1}\{Y_i^h = k\} \log l_k(\theta; \hat{Z}_i^l, X_i).$$

*over $\theta = (\alpha_1, \ldots, \alpha_K, \beta, \gamma)$. If the CoT prompting strategy is used to estimate $\mathbb{P}(Y_i^l = k \mid I_i, O_i)$, also assume Conditions C.1 and C.2 and let $n/m_n \to 0$ as $n \to \infty$. Then*

$$\sqrt{n}\big(\hat{\theta}_n - \theta^*\big) \xrightarrow{d} \mathcal{N}\big(0, \mathcal{I}(\theta^*)^{-1}\big) \quad and \quad \sqrt{n}\begin{pmatrix} \hat{\beta} - \beta^* \\ \hat{\gamma} - \gamma^* \end{pmatrix} \xrightarrow{d} \mathcal{N}\big(0, \{\mathcal{I}(\theta^*)^{-1}\}_{(\beta,\gamma)}\big) \quad as \ n \to \infty.$$

**Consistent variance estimator and confidence intervals.** To estimate the variance of $(\hat{\beta}, \hat{\gamma})$, calculate the observed Fisher information matrix: $\hat{\mathcal{I}}_{\text{obs}}(\hat{\theta}_n) = -(1/n)\nabla_\theta^2 \ell_n(\hat{\theta}_n; \hat{Z}^l, X)$ at the MLE $\hat{\theta}_n = (\hat{\alpha}_1, \ldots, \hat{\alpha}_K, \hat{\beta}, \hat{\gamma})$, where $\nabla_\theta^2 \ell_n$ is the second derivative matrix of $\ell_n$ with respect to all parameters $\theta = (\alpha_1, \ldots, \alpha_K, \beta, \gamma)$. Let $\hat{V} = \hat{\mathcal{I}}_{\text{obs}}(\hat{\theta}_n)^{-1}$. Extract the $(\beta, \gamma)$-block $\hat{V}_{(\beta,\gamma)}$, which is the bottom right $(1+d) \times (1+d)$ block of $\hat{V}$. The $100(1-\alpha)\%$ marginal confidence intervals (CIs) for the parameters are

$$\hat{\beta} \pm z_{1-\alpha/2}\frac{\sqrt{\hat{V}_{11}}}{\sqrt{n}}, \quad \hat{\gamma}_j \pm z_{1-\alpha/2}\frac{\sqrt{\hat{V}_{j+1,j+1}}}{\sqrt{n}}, \ j = 1, \ldots, d.$$

The joint confidence region is $\big\{(\beta, \gamma) : n[(\beta, \gamma) - (\hat{\beta}, \hat{\gamma})]\hat{V}_{(\beta,\gamma)}^{-1}[(\beta, \gamma) - (\hat{\beta}, \hat{\gamma})]^\top \le \chi_{d+1,1-\alpha}^2\big\}$. Under previous conditions, the variance estimator is consistent, and the coverage probability converges to $1 - \alpha$ as $n \to \infty$. Additionally, Appendix C.3 outlines methods for constructing confidence intervals for differentiable functions of $\theta$. These methods can be employed to construct prediction intervals and to evaluate the "partial effect" of a covariate, as further detailed in Appendix C.3.

## 4 Understanding and narrowing the human-LLM gap in practice

In this section, we examine the applications described in Section 3.2 using real-world data and popular LLM judges. We have included extra semi-synthetic (more controlled) and robustness-check experiments in Appendix B.

We begin by detailing the data used in our experiments.

### 4.1 LLM judges, datasets, and LLM judgments collection

**LLM judges.** We utilize six distinct LLM judges[6]: GPT-4.1 [32], GPT-4.1-nano [32], GPT-4o-mini [15], LLaMa-3.1-8B-Instruct [11], Selene-1-Mini [1], and Prometheus-v2 [18]. The GPTs and LLaMa-3.1-8B-It represent high-performing, general-purpose models, while Selene-1-Mini and Prometheus-v2 are specialized, state-of-the-art open judges.

**Datasets.** Our experiments are conducted using publicly available human-annotated datasets:

---

[6]The official model names are `gpt-4.1-nano`, `gpt-4.1`, `gpt-4o-mini-2024-07-18`, `meta-llama/Llama-3.1-8B-Instruct`, `AtlaAI/Selene-1-Mini-Llama-3.1-8B`, and `prometheus-eval/prometheus-8x7b-v2.0`.

- **BigGen Bench (BGB)**: BGB [17] evaluates language model outputs based on detailed rubrics across five satisfaction levels (originally from 1 to 5, converted here to 0 to 4). We focus on the subset containing English-language human annotations, comprising 695 instances across 77 tasks and nine evaluated capabilities (*e.g.*, planning, tool usage). Each instance has responses from four different models, totaling 2780 data points. We exclude a few data points with invalid annotations.

- **Chatbot Arena (CA)**: We use the dataset `arena-human-preference-100k` [40], derived from Chatbot Arena [6]. This dataset consists of 100k queries, each responded to simultaneously by two anonymous models, with user preferences annotated as either a clear choice or "good"/"bad" ties, where both responses are equilavently good or bad. We randomly select a subset of 5000 queries that are not multi-turn conversations and merge "good"/"bad" ties into a single category.

**Judgment collection.** Following Section 3, we gather LLM judgments via two distinct prompting methods. The first explicitly instructs the judges to provide only a rating without explanation, allowing us to extract log probabilities for each rating directly. The second prompts the judges to elaborate on their reasoning before explicitly stating their final judgment, from which we sample 50 times to estimate rating probabilities. We adapt distinct prompts for absolute and relative ratings: for absolute ratings, we modify reference-free Prometheus prompts from Kim et al. [18]; for relative ratings, we adjust AlpacaEval 2.0 [26] (for log probabilities) and ArenaHard prompts[7] [25] (for chain-of-thought reasoning). To ensure quality in our analyses and comparability across judges, we retain only instances where at least 25 valid CoT-generated scores were produced within 1k output tokens by all judges; this filtering removes at most 10% of instances per dataset. In our main experiments, we use log probabilities for closed models (GPTs) and chain-of-thought (CoT) sampling for other judges. We adopt CoT for open models since it generally yields higher-quality judgments, whereas for closed models, it is prohibitively costly to run, so we rely only on log probabilities. Prompt templates are provided in the Appendix A.

### 4.2 Application 1: Improved LLM judgements with few human annotations

In this application, our goal is to improve the performance of LLM judges using only a small number of human-annotated data points. This scenario is particularly relevant since (i) obtaining high-quality human annotations is costly, and (ii) fine-tuning an LLM judge becomes challenging when labeled examples are limited[8]. In this section, we demonstrate that our method can still provide benefits even without using covariates (*i.e.*, setting $\gamma = 0$), which can be hard to use when the training set is small.

**Metrics.** To evaluate judge quality, we consider three metrics: (i) cross-entropy loss, comparing the predicted rating probabilities with the actual labels; (ii) calibration error; and (iii) accuracy, which involves selecting the predicted class with the highest probability and comparing it with true labels. The results reported are averages across all judges. To compute the calibration error, we first calculate the error for each class individually and then average them. For class $k$, the steps for computing calibration error are: (i) using a probabilistic classifier (any of the methods reported in Figure 2), predict the probability of class $k$ for all $n_{te}$ test data points, obtaining $\{\hat{p}_{ki}\}_{i=1}^{n_{te}}$, (ii) discretize $\{\hat{p}_{ki}\}_{i=1}^{n_{te}}$ into 10 bins and, for each bin, compute the difference of the average predicted probability and the relative frequency of class $k$, and (iii) average these differences. To construct the bins, we use equally spaced quantiles of $\{\hat{p}_{ki}\}_{i=1}^{n_{te}}$.

**Data splitting.** Using a fixed random seed, we split each dataset into training and testing sets with an 80:20 ratio. For Chatbot Arena, we randomly divide data points into training and testing subsets. For BigGen Bench, we ensure instances do not overlap between the training and testing sets, simulating a realistic and challenging scenario in which new, unknown queries appear at test time. For a given sample size $n_{tr} \in \{20, 40, 80, 160, 320\}$, we randomly select $n_{tr}$ points from the full set of training queries to fit our models. Across all datasets, we perform this procedure using 10 different random seeds, and the reported results reflect averages and standard deviations across these splits.

**Methods.** We use two different versions of our method, assuming an ordinal structure of responses ("ordinal", default model defined in Section 3) or not ("multinomial", Appendix E). Regarding baselines, we primarily focus on two: the first baseline ("Raw") directly utilizes the raw probability

---

[7]ArenaHard has 5 levels: response A is much better than B, A is better than B, A and B are equally good, B is better than A, response B is much better than A. Given that we use three levels in our experiments, we compute the level probabilities estimates, and then convert to three levels by summing the probabilities of edge classes.

[8]Moreover, `Bridge` is still applicable in cases where only an inference API is provided.

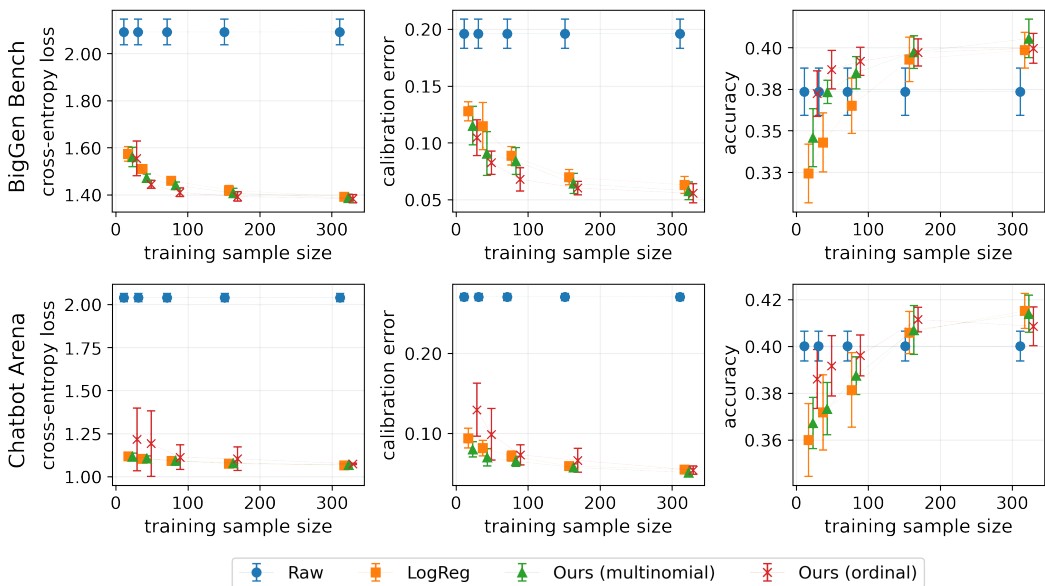

Figure 2: Performance comparison of our proposed methods, logistic-regression baseline, and raw LLM judgments across all datasets. Our methods consistently match or outperform the baselines, notably excelling on BigGen Bench, likely thanks to sensible inductive biases.

outputs from the LLM judges, while the second baseline ("LogReg") involves fitting a multinomial logistic regression model on top of these raw probabilities to potentially achieve better performance; this last baseline can be seen as a naive version of the "multinomial" approach but lacks a principled modeling foundation and is not interpretable. Additionally, we explore providing in-context learning (ICL) examples to the judge as another baseline, reporting results for this experiment in Appendix B; this last method performs poorly compared to other approaches.

**Results.** Figure 2 presents the experimental results. Across datasets, the different versions of our method and the logistic regression baseline outperform the raw LLM judgments consistently on all metrics. Notably, our methods never underperform relative to the baselines and significantly excel on the BigGen Bench. This advantage is likely due to the effective inductive biases provided by our models. Interestingly enough, the two different versions of our method perform well. However, the "ordinal" (default) method will often be preferable since it is simpler and more interpretable, as it explores the notion of order in the data and has fewer parameters.

### 4.3 Application 2: Detecting and testing for human-LLM gaps

Different from the previous experiment, this analysis incorporates a set of covariates $X$ that represent potential sources of discrepancies in LLM judgments relative to humans. We focus exclusively on covariates derived from the outputs $O$, as they are more direct to collect and interpret. Initially, we consider 47 interpretable covariates, comprising lightweight automated metrics (*e.g.*, word count, sentiment polarity) and features extracted via LLM scoring (*e.g.*, conciseness, fluency, creativity). For relative ratings, we compute differences between covariates from the second and first responses. We then cluster these covariates based on their correlations, substantially reducing their number by $\approx 30\%$ for BigGen Bench and $\approx 20\%$ for Chatbot Arena. Appendix D gives a comprehensive description of the used covariates, the clustering algorithm, and the resultant covariate clusters. After clustering, we extract the first principal component from each cluster and standardize the resulting variables to have zero mean and unit variance. Subsequently, we apply our proposed method, calculate p-values, and adjust them using the Benjamini-Yekutieli [2] procedure[9] for false discovery rate (FDR) control when conducting multiple tests at the same time.

**Results.** Tables 1 and 2 summarize our findings for BigGen Bench and Chatbot Arena, respectively. These tables include only covariates that show a statistically significant contribution for at least one judge; for full tables and unadjusted p-values, see Appendix B.6. The direction of these effects

---

[9]We use the Python package *statsmodels* for this adjustment: https://www.statsmodels.org/dev/generated/statsmodels.stats.multitest.multipletests.html

Table 1: Human-LLM judgement discrepancies on BigGen Bench

| | GPT-4.1-nano | GPT-4.1 | GPT-4o-mini | LLaMa-3.1-8B-It | Selene-1-Mini | Prometheus-v2 |
|---|---|---|---|---|---|---|
| Writing Quality | **-0.38***** | −0.10 | −0.02 | **-0.22**** | −0.02 | **-0.22***** |
| Text Length | **-0.83***** | **-0.39***** | **-0.43***** | **-0.78***** | **-0.44***** | **-0.74***** |
| Positive Sentiment | **-0.31***** | **-0.12*** | **-0.15**** | **-0.22**** | **-0.18***** | **-0.21***** |
| Layout Density | −0.23 | **-0.15*** | −0.11 | −0.21 | −0.13 | −0.17 |
| Causal Markers | **-0.19**** | −0.09 | −0.10 | −0.12 | −0.08 | −0.07 |
| Structure Counts | **0.35***** | **0.16*** | 0.11 | **0.29**** | 0.12 | **0.29***** |
| Sentiment | **0.24**** | 0.10 | 0.11 | 0.11 | 0.09 | 0.10 |
| Code Block | **0.20**** | 0.07 | 0.09 | **0.22***** | **0.14**** | **0.20***** |
| Character Density | 0.25 | 0.13 | 0.12 | 0.23 | **0.20**** | 0.13 |
| Compound Sentiment | **0.27***** | 0.06 | 0.08 | 0.16 | **0.13**** | **0.19**** |
| Question Count | 0.16 | 0.09 | **0.13*** | 0.13 | 0.11 | 0.15 |

Significance: *** $p < 0.01$, ** $p < 0.05$, * $p < 0.10$.

Table 2: Human-LLM judgement discrepancies on Chatbot Arena

| | GPT-4.1-nano | GPT-4.1 | GPT-4o-mini | LLaMa-3.1-8B-It | Selene-1-Mini | Prometheus-v2 |
|---|---|---|---|---|---|---|
| Text Length | **-2.05***** | **-0.54***** | **-1.02***** | **-1.61**** | **-1.17**** | **-1.20***** |
| Creativity/Engagement | **-1.27***** | **-0.32***** | **-0.64***** | **-1.10**** | **-0.78**** | **-0.77***** |
| Bold Text | **0.74**** | **0.25***** | **0.50***** | **0.77**** | **0.66***** | **0.62***** |

Significance: *** $p < 0.01$, ** $p < 0.05$, * $p < 0.10$.

is important: positive values indicate attributes preferred more strongly by LLM judges, whereas negative values indicate attributes preferred by humans. Some insights about the results are:

- Across datasets and judges, longer responses receive systematically lower scores, showing that LLM judges favor brevity relative to humans. This contrasts with Dubois et al. [8], who argue that length-controlled (assuming LLMs are positively biased towards lengthier responses) scoring aids alignment. In Appendix B.6, we show that GPT-4-Turbo (main judge on AlpacaEval) exhibits the same pattern. The discrepancy with Dubois et al. [8] likely stems from methodological differences, our instance-level analysis versus their system-level aggregation, which introduces extra complications to comparability. At the end of the day, working on the instance level is a more direct and reliable way of drawing such conclusions.

- Human annotators reward creativity and engaging responses more than LLM judges, a discrepancy most pronounced on Chatbot Arena. This pattern is intuitive: users of that platform often are there to "play" with generations, while the LLM judges were never instructed to value creativity. A similar conclusion can be drawn from the "Positive sentiment" dimension in BigGen Bench.

- Bias profiles overlap considerably across LLM judges, suggesting common underlying biases that are inherited from similar training sets and procedures.

In Appendix B.6, we conduct extra related analyses. For example, we check how robust our findings are for different sample sizes. We split Chatbot Arena queries into technical and non-technical categories using GPT-4o-mini as a zero-shot classifier. For technical queries, HumanLLM divergences were not statistically significant, suggesting that discrepancies arise mainly

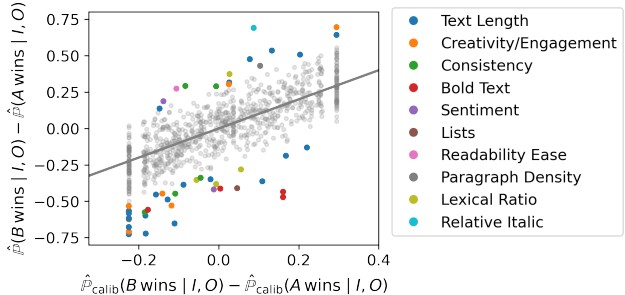

Figure 3: Covariates $X$ are important. Dots indicate how adding covariates alters the predicted human preference, with colors marking the most influential.

in subjective, non-technical content. An important limitation of this analysis is the reduced sample size after splitting, which lowers statistical power.

To quantify the practical impact of the bias covariates, we zoom in on the Chatbot Arena evaluation with the Selene-1-Mini judge. Two variants of our model are fitted: the full specification, denoted $\hat{\mathbb{P}}$, which incorporates the gap term $\gamma^\top X$, and a "calibration" variant, $\hat{\mathbb{P}}_{\text{calib}}$, obtained by setting $\gamma = 0$. For every pair of responses $(O_A, O_B)$ and variants of our method, we predict the probabilities that humans prefer $B$ over $A$ and $A$ over $B$ and then take their difference. Figure 3 plots these differences, highlighting instances where biasing covariates substantially alter the prediction. Each point is colored by the covariate whose contribution $|\gamma_j X_{ij}|$ is largest in magnitude, thereby revealing the dominant factor driving each discrepancy. The discrepancies have large magnitudes for some data points. As extra results, we place figures in Appendix B.6 that show that including covariates in our model can induce improved prediction performance, even though the biggest improvements are obtained without the need to add extra covariates.

## 5 Discussion

We propose `Bridge`, a unified statistical framework that simultaneously models ratings from both human annotators and LLM judges. The framework couples a statistical model with a specialized estimation procedure, enabling (i) principled calibration/limitations of LLM scores and (ii) a clearer characterization of the divergences between human and LLM evaluations.

**Limitations.** The chief limitation of our approach is vulnerability to model misspecification. When the assumed data-generating process is inaccurate, owing to unrealistic distributional assumptions or omitted covariates, the resulting parameter estimates must be interpreted cautiously; please check Appendix B.3 for a detailed discussion on model misspecification. A second practical challenge is the construction of informative covariates $X$; while users can start with generic, off-the-shelf metrics (e.g., response length, readability grade), domain knowledge could be needed to devise variables that capture salient sources of differences between LLM and human judgments, especially when those also depend on the input $I$ and not only on the output $O$.

**The significance of this work in today's LLM-evaluation landscape.** We recognize that as LLM tasks become more sophisticated, achieving a truly reliable "gold standard" through human annotation is increasingly difficult. However, `Bridge` is flexible by design and is not tied to a single definition of the gold standard. Whether the benchmark consists of individual human judgments, a consensus among multiple annotators, or even alternative proxies, `Bridge` is meant to detect and reduce inconsistencies between whatever reference is chosen and LLM assessments. Moreover, aligning LLMs with human preferences and judgments will continue to be important for many real-world applications, especially where trust, safety, and social acceptance matter.

**Observational vs. experimental data.** Our experiments rely exclusively on observational data, in the sense that we do not intervene on $X$. This choice has both strengths and limitations. On the one hand, observational data capture the diversity and unpredictability of real user-LLM interactions, enhancing the external validity of our findings. On the other hand, because the data are not generated under controlled interventions, our estimated parameters should be interpreted as descriptive associations rather than causal effects. Confounding variables and selection biases may also influence these relationships.

**Future work and extensions.** Promising directions include developing estimation routines robust to model misspecification, extending the framework to automatically infer divergence factors, and leveraging representation learning to construct covariates $X$ on the fly. In principle, `Bridge` can also incorporate text embeddings as $X$ (provided enough training data and regularization), though we currently see no clear advantage in doing so; nevertheless, this remains an interesting avenue for future exploration. Extending `Bridge` to open-ended, natural-language evaluations is an important direction for future work and will likely require principled ways to represent free-form judgments as comparable quantities. Moreover, `Bridge` can also be applied in settings where model outputs span multiple modalities (*e.g.*, image, video, audio). In such cases, however, constructing meaningful covariates $X$ may be more challenging. We encourage adoption and extensions of `Bridge` in these directions for future work.

## 6 Acknowledgements

This paper is supported by the National Science Foundation (NSF) grants DMS-2027737, DMS-2113373, DMS-2414918, SES-1846747, and a gift from OpenAI.

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

# A Prompt templates

## A.1 BigGen Bench prompts (absolute ratings)

For BigGen Bench, the system prompt is instance-dependent. Therefore, we do not report them here.

---

**BigGen Bench (logprobs)**

```
###Task Description:
An instruction (might include an Input inside it), a response
to evaluate, and a score rubric representing a evaluation
criteria are given.
1. Write a score that is an integer between 1 and 5. You should
refer to the score rubric.
2. Your output must be only an integer number between 1 and 5,
and nothing else
3. Please do not generate any other opening, closing, and
explanations. Do not include any spaces or linebreaks before
your judgement."

###The instruction to evaluate:
{instruction}

###Response to evaluate:
{response}

###Score Rubrics:
{rubric}

###Score:
```

---

**BigGen Bench (CoT)**

```
###Task Description:
An instruction (might include an Input inside it), a response
to evaluate, and a score rubric representing a evaluation
criteria are given.
1. Write a detailed feedback that assess the quality of the
response strictly based on the given score rubric, not
evaluating in general.
2. After writing a feedback, write a score that is an integer
between 1 and 5. You should refer to the score rubric.
3. The output format should look as follows: "(write a feedback
for criteria) [RESULT] (an integer number between 1 and 5)"
4. Please do not generate any other opening, closing, and
explanations.

###The instruction to evaluate:
{instruction}

###Response to evaluate:
{response}

###Score Rubrics:
{rubric}

###Feedback:
```

## A.2 Chatbot Arena prompts (relative ratings)

---

**Chatbot Arena system prompt (logprobs)**

```
You are a highly efficient assistant, who evaluates and rank
large language models (LLMs) based on the quality of their
responses to given prompts. This process will create a
leaderboard reflecting the most accurate and human-preferred
answers.
```

---

**Chatbot Arena user prompt (logprobs)**

```
I require a leaderboard for various large language models. I'll
provide you with prompts given to these models and their
corresponding outputs. Your task is to assess these responses,
and select the model that produces the best output from a human
perspective. The input prompt can possibly include an image; in
that case the user question or instruction will be related to
that image and you must take that into account.

## Instruction

{
    "instruction": "{instruction}",
}

## Model Outputs

Here are the unordered outputs from the models. Each output is
associated with a specific model, identified by a unique model
identifier.

{
    {
        "model_identifier": "A",
        "output": "{output_1}"
    },
    {
        "model_identifier": "B",
        "output": "{output_2}"
    }
}

## Task

Evaluate the models based on the quality and relevance of their
outputs, and be prepared for the possibility of a tie. If one
model clearly produces the best output, respond with its
identifier. However, if the responses are equally good or bad
and result in a tie, return C. Your output must contain only
one of these identifiers (no quotes, spaces, or new lines): A,
B, or C.

## Judgment

Best Model Identifier:
```

## Chatbot Arena system prompt (CoT)

```
Please act as an impartial judge and evaluate the quality of
the responses provided by two AI assistants to the user prompt
displayed below. The input prompt can possibly include an
image; in that case the user question or instruction will be
related to that image and you must take that into account. You
will be given assistant A's answer and assistant B's answer.
Your job is to evaluate which assistant's answer is better.

Begin your evaluation by generating your own answer to the
prompt. You must provide your answers before judging any
answers.

When evaluating the assistants' answers, compare both
assistants' answers with your answer. You must identify and
correct any mistakes or inaccurate information.

Then consider if the assistant's answers are helpful, relevant,
and concise. Helpful means the answer correctly responds to the
prompt or follows the instructions. Note when user prompt has
any ambiguity or more than one interpretation, it is more
helpful and appropriate to ask for clarifications or more
information from the user than providing an answer based on
assumptions. Relevant means all parts of the response closely
connect or are appropriate to what is being asked. Concise
means the response is clear and not verbose or excessive.

Then consider the creativity and novelty of the assistant's
answers when needed. Finally, identify any missing important
information in the assistants' answers that would be beneficial
to include when responding to the user prompt.

After providing your explanation, you must output only one of
the following choices as your final verdict with a label:

1. Assistant A is significantly better: [[A>>B]]
2. Assistant A is slightly better: [[A>B]]
3. Tie, relatively the same: [[A=B]]
4. Assistant B is slightly better: [[B>A]]
5. Assistant B is significantly better: [[B>>A]]

Example output: "My final verdict is tie: [[A=B]]".
```

## Chatbot Arena user prompt (CoT)

```
<|User Prompt|>
{instruction}

<|The Start of Assistant A's Answer|>
{output_1}
<|The End of Assistant A's Answer|>

<|The Start of Assistant B's Answer|>
{output_2}
<|The End of Assistant B's Answer|>
```

### A.3 Prompts for extracting LLM-scored covariates

The following prompts were used to obtain LLM-scored covariates, adapted from the prompts by [28].

---

**Coherence**

```
You will be given one LLM response, generated in reply to a
human prompt. Note that only the LLM response is provided for
evaluation.

Your task is to rate the LLM response on one evaluation metric.

Please make sure you read and understand these instructions
carefully. Please keep this document open while reviewing, and
refer to it as needed.

Evaluation Criteria:
Coherence (0-5): Assess how well the LLM response is structured
and organized.
A highly coherent response (score 5) will present ideas in a
clear, logical progression. The text should flow naturally,
with each sentence and paragraph connecting logically to build
a coherent narrative or argument.
A score of 0 indicates a response that is disorganized,
disconnected, or otherwise hard to follow.

Evaluation Steps:
1. Read the LLM response carefully to understand its content
and structure.
2. Assess overall structure and flow. Determine whether the
response is well-organized and whether the ideas and arguments
progress in a logical order.
3. Assign a score for coherence on a scale of 0 to 5, where 0
is the lowest and 5 is the highest based on the Evaluation
Criteria.
Provide only a single numeric value (e.g., 0.75) without any
additional text.

Response:
{response}
```

---

**Factuality**

```
You will be given one LLM response, generated in reply to a
human prompt. Note that only the LLM response is provided for
evaluation.

Your task is to rate the LLM response on one evaluation metric.

Please make sure you read and understand these instructions
carefully. Please keep this document open while reviewing, and
refer to it as needed.

Evaluation Criteria:
Factuality (0-5): Assess how factually accurate and
evidence-based the LLM response is. A highly factual response
(score 5) will contain accurate, verifiable information. A
score of 0 indicates a response that includes inaccuracies or
unsupported claims.
```

```
Evaluation Steps:
1. Read the LLM response carefully to identify its content.
2. Check if the response contains information that can be
verified as accurate.
3. Assign a score for factuality on a scale of 0 to 5, where 0
is the lowest and 5 is the highest based on the Evaluation
Criteria.
Provide only a single numeric value (e.g., 0.75) without any
additional text.

Response:
{response}
```

## Clarity

```
You will be given one LLM response, generated in reply to a
human prompt. Note that only the LLM response is provided for
evaluation.

Your task is to rate the LLM response on one evaluation metric.

Please make sure you read and understand these instructions
carefully. Please keep this document open while reviewing, and
refer to it as needed.

Evaluation Criteria:
Clarity (0-5): Assess how clear and understandable the language
of the LLM response is. A highly clear response (score 5) will
communicate ideas in a straightforward and unambiguous manner.
A score of 0 indicates a response that is vague or confusing.

Evaluation Steps:
1. Read the LLM response carefully to understand its content
and intent.
2. Evaluate whether the language used is precise and easy to
follow.
3. Assign a score for clarity on a scale of 0 to 5, where 0 is
the lowest and 5 is the highest based on the Evaluation
Criteria.
Provide only a single numeric value (e.g., 0.75) without any
additional text.

Response:
{response}
```

## Conciseness

```
You will be given one LLM response, generated in reply to a
human prompt. Note that only the LLM response is provided for
evaluation.

Your task is to rate the LLM response on one evaluation metric.

Please make sure you read and understand these instructions
carefully. Please keep this document open while reviewing, and
refer to it as needed.
```

```
Evaluation Criteria:
Conciseness (0-5): Assess how succinct and to-the-point the LLM
response is. A highly concise response (score 5) will deliver
its message without unnecessary verbosity. A score of 0
indicates a response that is overly wordy or includes redundant
details.

Evaluation Steps:
1. Read the LLM response carefully to capture its core ideas.
2. Evaluate whether the response expresses its content in a
succinct manner.
3. Assign a score for conciseness on a scale of 0 to 5, where 0
is the lowest and 5 is the highest based on the Evaluation
Criteria.

Provide only a single numeric value (e.g., 0.75) without any
additional text.

Response:
{response}
```

## Creativity

```
You will be given one LLM response, generated in reply to a
human prompt. Note that only the LLM response is provided for
evaluation.

Your task is to rate the LLM response on one evaluation metric.

Please make sure you read and understand these instructions
carefully. Please keep this document open while reviewing, and
refer to it as needed.

Evaluation Criteria:
Creativity (0-5): Assess how original and inventive the LLM
response is. A highly creative response (score 5) will present
ideas in a unique and engaging way. A score of 0 indicates a
response that is unoriginal or formulaic.

Evaluation Steps:
1. Read the LLM response carefully to appreciate its content
and style.
2. Evaluate whether the response demonstrates inventive thought
and originality in its presentation.
3. Assign a score for creativity on a scale of 0 to 5, where 0
is the lowest and 5 is the highest based on the Evaluation
Criteria.
Provide only a single numeric value (e.g., 0.75) without any
additional text.

Response:
{response}
```

## Consistency

```
You will be given one LLM response, generated in reply to a
human prompt. Note that only the LLM response is provided for
evaluation.
```

```
Your task is to rate the LLM response on one evaluation metric.

Please make sure you read and understand these instructions
carefully. Please keep this document open while reviewing, and
refer to it as needed.

Evaluation Criteria:
Consistency (0-5): Assess how uniform and steady the style and
content of the LLM response are. A highly consistent response
(score 5) will maintain a uniform approach throughout without
contradictions. A score of 0 indicates a response that contains
conflicting information or fluctuates in style.

Evaluation Steps:
1. Read the LLM response carefully to understand its content
and style.
2. Evaluate whether the response maintains consistency in its
presentation.
3. Assign a score for consistency on a scale of 0 to 5, where 0
is the lowest and 5 is the highest based on the Evaluation
Criteria.
Provide only a single numeric value (e.g., 0.75) without any
additional text.

Response:
{response}
```

## Engagement

```
You will be given one LLM response, generated in reply to a
human prompt. Note that only the LLM response is provided for
evaluation.

Your task is to rate the LLM response on one evaluation metric.

Please make sure you read and understand these instructions
carefully. Please keep this document open while reviewing, and
refer to it as needed.

Evaluation Criteria:
Engagement (0-5): Assess how engaging the LLM response is to a
reader. A highly engaging response (score 5) will capture and
sustain the reader's attention effectively. A score of 0
indicates a response that is dull or fails to hold interest.

Evaluation Steps:
1. Read the LLM response carefully to understand its content
and appeal.
2. Evaluate whether the response is able to maintain a reader's
interest throughout.
3. Assign a score for engagement on a scale of 0 to 5, where 0
is the lowest and 5 is the highest based on the Evaluation
Criteria.
Provide only a single numeric value (e.g., 0.75) without any
additional text.

Response:
{response}
```

## Fluency

You will be given one LLM response, generated in reply to a human prompt. Note that only the LLM response is provided for evaluation.

Your task is to rate the LLM response on one evaluation metric.

Please make sure you read and understand these instructions carefully. Please keep this document open while reviewing, and refer to it as needed.

Evaluation Criteria:
Fluency (0-5): Assess how smoothly and naturally the LLM response reads. A highly fluent response (score 5) will have a natural flow and is easy to read and follow. A score of 0 indicates a response that is choppy or awkward in its language, or is hard to understand.

Evaluation Steps:
1. Read the LLM response carefully to understand its content and structure.
2. Evaluate whether the response exhibits a smooth and natural flow of language.
3. Assign a score for fluency on a scale of 0 to 5, where 0 is the lowest and 5 is the highest based on the Evaluation Criteria.
Provide only a single numeric value (e.g., 0.75) without any additional text.

Response:
{response}

## Appropriateness

You will be given one LLM response, generated in reply to a human prompt. Note that only the LLM response is provided for evaluation.

Your task is to rate the LLM response on one evaluation metric.

Please make sure you read and understand these instructions carefully. Please keep this document open while reviewing, and refer to it as needed.

Evaluation Criteria:
Appropriateness (0-5): Assess whether the tone and style of the LLM response are suitable for the intended content. An appropriate response (score 5) will use language and tone that fits the content and purpose. A score of 0 indicates a response that is mismatched in tone or style for the given context.

Evaluation Steps:
1. Read the LLM response carefully to understand its content.
2. Evaluate whether the tone and style are appropriate for the intended content and context.
3. Assign a score for appropriateness on a scale of 0 to 5, where 0 is the lowest and 5 is the highest based on the Evaluation Criteria.

```
Provide only a single numeric value (e.g., 0.75) without any
additional text.

Response:
{response}
```

## Sentiment

```
You will be given one LLM response, generated in reply to a
human prompt. Note that only the LLM response is provided for
evaluation.

Your task is to rate the LLM response on one evaluation metric.

Please make sure you read and understand these instructions
carefully. Please keep this document open while reviewing, and
refer to it as needed.

Evaluation Criteria:
Sentiment (0-5): Assess the overall emotional tone of the LLM
response. A response with a highly positive sentiment (score 5)
will convey optimism and positive emotion, while a score of 0
indicates a negative sentiment that conveys negative emotion.

Evaluation Steps:
1. Read the LLM response carefully to understand its emotional
undertone.
2. Evaluate whether the response expresses a positive emotional
tone.
3. Assign a score for sentiment on a scale of 0 to 5, where 0
is the lowest (not positive) and 5 is the highest (positive)
based on the Evaluation Criteria.
Provide only a single numeric value (e.g., 0.75) without any
additional text.

Response:
{response}
```

# B Additional empirical results

## B.1 Probability Distribution Reconstruction Loss via Logit Trick

Figures 4 and 5 report the values of

$$\min_{\substack{\bar{\eta}_1 < \cdots < \bar{\eta}_K \in \mathbb{R}, \\ z_1, \ldots, z_n \in \mathbb{R}}} \frac{1}{nK} \sum_{i=1}^{n} \sum_{k=0}^{K} \left| p_k(\bar{\eta}_1, \ldots, \bar{\eta}_K, z_i) - \mathbb{P}(Y_i^l = k \mid I_i, O_i) \right|,$$

which is used in the logit trick. This metric provides a measure of how well the ordinal assumption captures LLM judgments. From the figures, we observe that the average deviation is below $0.04$ for all judges. Deviations are typically slightly higher for judges whose judgment probabilities we estimate, reflecting finite-sample approximations.

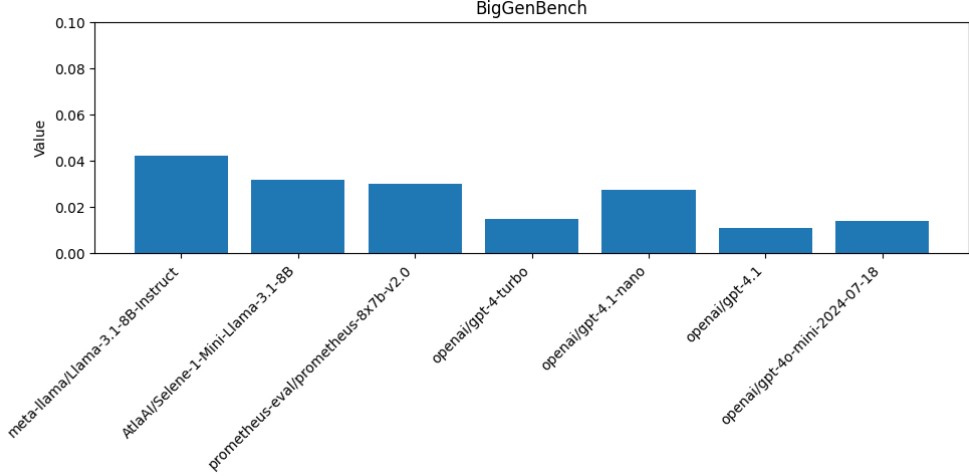

Figure 4: Reconstruction loss for BigGenBench

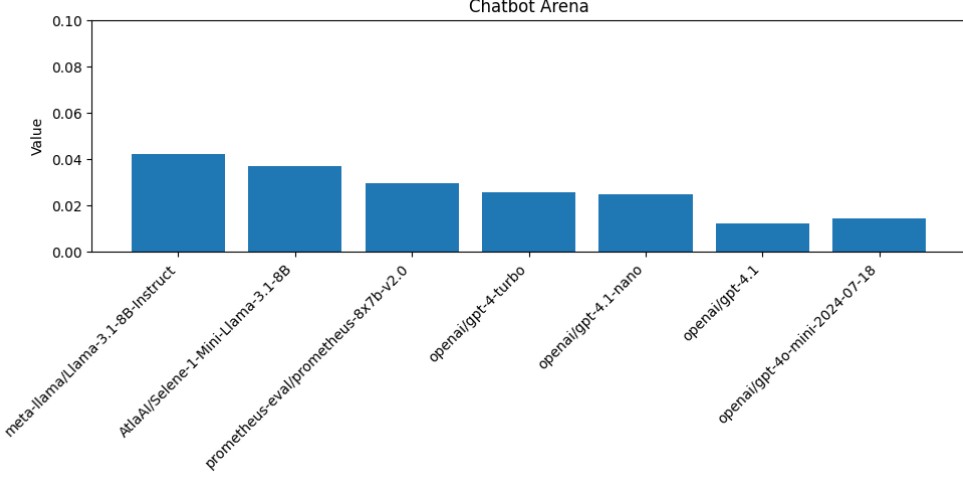

Figure 5: Reconstruction loss for Chatbot Arena

## B.2 Controlled experiments to show the effectiveness of the framework

We have conducted two additional controlled experiments to further validate our method, both of which use semi-synthetic setups that are more realistic than purely artificial data.

**First experiment:** We use GPT-4o-mini to simulate human ratings on BigGenBench queries. Next, we run GPT-4o-mini again, this time artificially biasing its latent scores $Z^l$ to *disfavor* specific markdown features; namely, bold/italicized words, headers, and lists. For each markdown feature (corresponding to the covariates $X_j$, $j = 1, 2, 3$), we bias $Z^l$ by subtracting $X_j$ (one at a time). We then estimate $\gamma = (\gamma_1, \gamma_2, \gamma_3)$ with our method. In this controlled setting, biasing toward feature $j$ should result in $\gamma_j = -1$ while $\gamma_i = 0$ for $i \neq j$. Standard errors are shown in parentheses.

Table 3: Estimated bias parameters $\gamma$ (with standard errors) in the tightly controlled experiment.

| Setting | $\gamma_1$ (SE) | $\gamma_2$ (SE) | $\gamma_3$ (SE) |
|---|---|---|---|
| no bias | -0.26 (0.21) | -0.36 (0.35) | -0.17 (0.16) |
| bold/italic | -1.26 (0.21) | -0.36 (0.35) | -0.17 (0.16) |
| headers | -0.26 (0.21) | -1.36 (0.35) | -0.17 (0.16) |
| lists | -0.26 (0.21) | -0.36 (0.35) | -1.17 (0.16) |

**Second experiment:** We conduct a slightly less controlled experiment by prompting GPT-4o-mini to give lower scores to responses containing each markdown feature, one at a time. Since this manipulation is done through prompting rather than direct latent score adjustment, the resulting biases are not as clean; for example, the LLM tends to be consistently biased against lists. Still, the results largely follow the expected direction.

Table 4: Estimated bias parameters $\gamma$ (with standard errors) in the prompt-based experiment.

| Setting | $\gamma_1$ (SE) | $\gamma_2$ (SE) | $\gamma_3$ (SE) |
|---|---|---|---|
| no bias | -0.255 (0.208) | -0.362 (0.345) | -0.167 (0.161) |
| bold/italic | -1.992 (0.290) | 0.466 (0.486) | -1.583 (0.234) |
| headers | -0.069 (0.427) | -1.014 (0.713) | -3.061 (0.352) |
| lists | -0.172 (0.319) | -0.229 (0.536) | -5.660 (0.260) |

These experiments demonstrate that our framework can recover the direction and magnitude of induced discrepancies/biases, both in tightly controlled and more realistic, prompt-based scenarios.

### B.3   Robustness against model misspecification

When our model is used for prediction, misspecification is not a significant concern; much of machine learning relies on models that are not exactly correct. If our focus is on statistical inference, the model can still be valuable for uncovering discrepancies in LLM judgments, provided the misspecification is not too severe. Moreover, the linear predictor we use is quite flexible, as we can include any basis functions of $X$ as covariates (and then capture nonlinear relationships).

Empirically, we present below a simple simulation in which we introduce a mild nonlinearity into the LLM's latent score generation to test robustness to misspecification, both for prediction and inference. We draw $Z_i^h$ from a Normal distribution $\mathcal{N}(0, 1)$ and sample $Y_i^h$ from Categorical$(p(\alpha, Z_i^h))$. We then set

$$Z_i^l = \beta Z_i^h + \gamma^\top X_i + \delta(\gamma^\top X_i)^2,$$

where $X_i$ is drawn from a multivariate Normal $\mathcal{N}(0, I_3)$ distribution and $\delta$ takes values in $\{0, 0.1, 0.25, 0.5, 1, 5\}$, controlling the degree of quadratic distortion. We set $\beta = 1$, $\gamma = (1, 1, 1)$, and $\alpha = (-1, 1)$. LLM judgments $Y_i^l$ are sampled from the usual ordered-logit link $p(\eta, Z_i^l)$, and we fit our original linear model, assuming $Z_i^l = \beta Z_i^h + \gamma^\top X_i$. By comparing the estimated parameters $(\hat{\beta}, \hat{\gamma}, \hat{Z}^h, \mathbb{P}(Y^h = k \mid I, O))$ to their true values in terms of mean absolute error (MAE) as $\delta$ increases, we directly measure the impact of model misspecification. The results below (Table 5) show that we can still recover $\gamma$ (then at least we know approximately how big are the main effects; main channel of discrepancies) and predict $\mathbb{P}(Y^h = k \mid I, O)$ with high accuracy, even under moderate

misspecification. We will add this experiment to the paper. Interestingly, the most affected results are the ones for $\beta$ and $Z^h$, which is of less interest.

Table 5: Mean absolute error (MAE) of parameter estimations.

| $\delta$ | $\beta$ | $\gamma$ | $Z^h$ | $\mathbb{P}(Y^h = k \mid I, O)$ |
|---|---|---|---|---|
| 0 | 0.010 | 0.014 | 0.014 | 0.002 |
| 0.1 | 0.024 | 0.014 | 0.072 | 0.010 |
| 0.25 | 0.047 | 0.016 | 0.169 | 0.025 |
| 0.5 | 0.266 | 0.017 | 0.340 | 0.045 |
| 1 | 0.960 | 0.011 | 0.563 | 0.077 |
| 5 | 24.958 | 0.345 | 0.789 | 0.117 |

## B.4 For Application 2, how do results vary based on the amount of training data?

To assess the robustness of our method in detecting human-LLM gaps, we repeat our analysis using varying random fractions of the available data. We set a significance threshold of 10% (i.e., $p$-values below 0.10 are considered significant) and, for each LLM, treat the detection of nonzero $\gamma_j$ coefficients as a binary classification problem (reject vs. not reject the null hypothesis). The "ground truth" label is determined using the full dataset. For each data fraction, we evaluate how well the method predicts these significance decisions by reporting precision, recall, and accuracy. Our results (see tables below) show that, for both BigGenBench and Chatbot Arena, strong precision and accuracy can be achieved with as little as 50% of the data. Recall is more challenging to improve, indicating that some discrepancies may require more data to detect reliably. For this experiment, we do not correct the p-values using the B-Y procedure.

Table 6: Significance prediction performance for BigGenBench as a function of training data fraction.

| % Data | Precision | Recall | Accuracy |
|---|---|---|---|
| 10 | 0.67 | 0.14 | 0.56 |
| 25 | 0.63 | 0.23 | 0.57 |
| 50 | 0.90 | 0.60 | 0.78 |
| 75 | 0.90 | 0.79 | 0.86 |
| 100 | 1.00 | 1.00 | 1.00 |

Table 7: Significance prediction performance for Chatbot Arena as a function of training data fraction.

| % Data | Precision | Recall | Accuracy |
|---|---|---|---|
| 10 | 0.33 | 0.08 | 0.75 |
| 25 | 0.92 | 0.38 | 0.85 |
| 50 | 0.61 | 0.46 | 0.80 |
| 75 | 0.93 | 0.69 | 0.92 |
| 100 | 1.00 | 1.00 | 1.00 |

Overall, these results suggest that our method for detecting human-LLM discrepancies is quite robust, with high precision and accuracy even when only half of the data is used. Recall improves with larger data fractions, highlighting the benefit of more data for sensitivity to weaker effects.

## B.5 Application 1

### B.5.1 ICL baseline

We conduct an additional experiment where we provide in-context learning (ICL) examples to the judge, using the same training samples employed by our method, to assess whether this strategy improves performance. Given the high token count, context length constraints, and associated computational costs, we limit this experiment to a single random split from Section 4.2, use only GPT4o-mini, and cap the number of training examples at 80. Figure 6 compares ICL with raw LLM scores, the logistic regression baseline, and our default "ordinal" method. While ICL yields slight improvements, its benefits remain marginal relative to our approach.

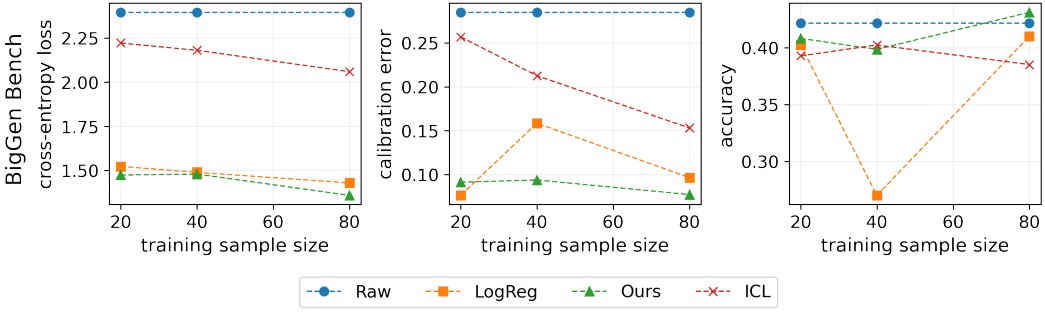

Figure 6: In-context-learning judge provides only marginal gains

## B.6    Application 2

### B.6.1    Performance gains

In this section, we show that including covariates in the model can lead to some performance gains if the objective is human alignment. In the next figures, we compare raw LLM judgements ("raw") with the application of our method using covariates ("covs") or not ("calib"). For both BGB and CA, we have gains in terms of cross-entropy loss (Figure 7), giving hints of better human-aligned judgments, while the gains in accuracy are only more pronounced in CA judgments (Figure 9).

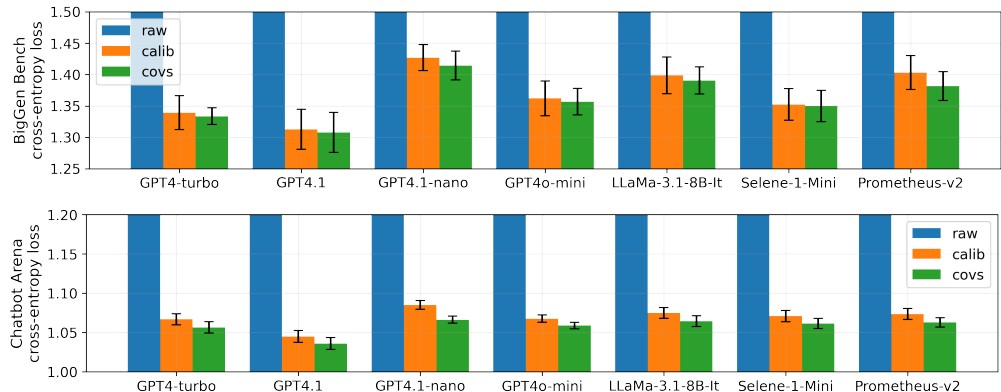

Figure 7: Performance in terms of cross-entropy loss.

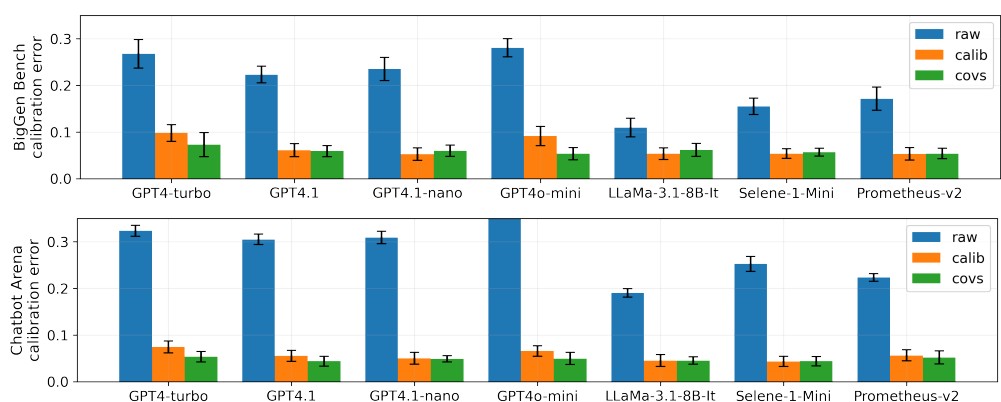

Figure 8: Performance in terms of probabilistic calibration.

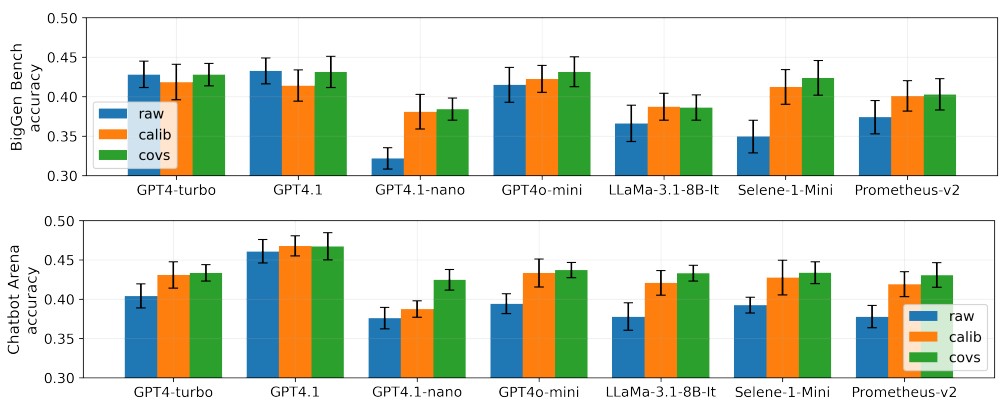

Figure 9: Performance in terms of accuracy.

### B.6.2 Tables BigGen Bench

Table 8: Human-LLM judgement discrepancies on BigGen Bench (with *no* Benjamini-Yekutieli correction)

| | GPT4-turbo | GPT4.1-nano | GPT4.1 | GPT4o-mini | LLaMa-3.1-8B-It | Selene-1-Mini | Prometheus-v2 |
|---|---|---|---|---|---|---|---|
| Writing Quality | **-0.07*** | **-0.38*** | **-0.10*** | −0.02 | **-0.22*** | −0.02 | **-0.22*** |
| Text Length | **-0.49*** | **-0.83*** | **-0.39*** | **-0.43*** | **-0.78*** | **-0.44*** | **-0.74*** |
| Italics | 0.04 | 0.06 | 0.05 | 0.04 | **0.08*** | 0.05 | 0.06 |
| Bold Text | −0.05 | −0.04 | 0.01 | −0.04 | −0.08 | −0.04 | −0.06 |
| Lists | 0.07 | 0.07 | 0.04 | **0.08*** | 0.09 | **0.11**** | 0.05 |
| Headers | 0.03 | 0.02 | 0.03 | 0.01 | −0.01 | 0.01 | 0.03 |
| Creativity/Engagement | 0.07 | 0.04 | 0.05 | **0.09**** | 0.06 | **0.09*** | 0.02 |
| Positive Sentiment | **-0.18*** | **-0.31*** | **-0.12*** | **-0.15*** | **-0.22*** | **-0.18*** | **-0.21*** |
| Conciseness | −0.07 | −0.09 | **-0.07*** | −0.02 | −0.07 | −0.04 | −0.02 |
| Contrast Markers | 0.01 | 0.00 | 0.02 | −0.01 | 0.02 | −0.01 | −0.01 |
| Layout Density | **-0.10*** | **-0.23**** | **-0.15*** | **-0.11*** | **-0.21**** | **-0.13**** | **-0.17**** |
| Causal Markers | **-0.13*** | **-0.19*** | **-0.09*** | **-0.10*** | **-0.12**** | **-0.08**** | −0.07 |
| Structure Counts | **0.15**** | **0.35*** | **0.16*** | **0.11*** | **0.29*** | **0.12**** | **0.29*** |
| Sentiment | **0.12*** | **0.24*** | **0.10**** | **0.11*** | 0.11 | **0.09*** | 0.10 |
| Readability Grade | 0.03 | 0.24 | 0.13 | −0.01 | **0.27*** | 0.07 | **0.27*** |
| Exclamation Density | −0.05 | **-0.18**** | −0.08 | −0.06 | −0.01 | 0.01 | −0.02 |
| Readability Ease | 0.10 | **0.39**** | **0.18*** | 0.08 | **0.45**** | 0.17 | **0.43*** |
| Polarity | 0.01 | −0.04 | 0.02 | 0.00 | −0.01 | 0.01 | 0.01 |
| Question Density | **-0.10**** | −0.09 | −0.06 | **-0.11**** | **-0.11*** | **-0.09*** | **-0.12**** |
| Paragraph Length | 0.02 | 0.04 | 0.01 | 0.04 | 0.05 | 0.03 | 0.07 |
| Code Block | **0.08**** | **0.20*** | **0.07**** | **0.09**** | **0.22*** | **0.14*** | **0.20*** |
| Additive Markers | −0.01 | 0.05 | 0.00 | 0.00 | 0.06 | 0.00 | 0.03 |
| Summary Markers | **-0.08**** | **-0.09*** | **-0.08*** | **-0.08**** | **-0.11**** | **-0.10*** | **-0.12**** |
| Character Density | **0.10*** | **0.25**** | **0.13**** | **0.12**** | **0.23**** | **0.20*** | 0.13 |
| Exclamation Count | 0.03 | **0.13*** | 0.03 | −0.01 | 0.08 | −0.03 | 0.05 |
| Lexical Ratio | −0.04 | −0.10 | 0.05 | −0.02 | **-0.20**** | −0.08 | **-0.13*** |
| Subjectivity | 0.00 | 0.02 | 0.00 | 0.02 | 0.02 | 0.01 | 0.02 |
| Sentence Length | −0.04 | **-0.13**** | **-0.07*** | −0.03 | **-0.10**** | −0.04 | **-0.10**** |
| Example Markers | 0.04 | 0.05 | 0.02 | 0.02 | 0.06 | 0.02 | 0.04 |
| Compound Sentiment | **0.11*** | **0.27*** | **0.06*** | **0.08*** | **0.16*** | **0.13*** | **0.19*** |
| Question Count | **0.12*** | **0.16*** | **0.09*** | **0.13*** | **0.13**** | **0.11*** | **0.15*** |

Significance: *** $p < 0.01$, ** $p < 0.05$, * $p < 0.10$.

Table 9: Human-LLM judgement discrepancies on BigGen Bench (with Benjamini-Yekutieli correction)

| | GPT4-turbo | GPT4.1-nano | GPT4.1 | GPT4o-mini | LLaMa-3.1-8B-It | Selene-1-Mini | Prometheus-v2 |
|---|---|---|---|---|---|---|---|
| Writing Quality | −0.07 | **-0.38\*\*\*** | −0.10 | −0.02 | **-0.22\*\*** | −0.02 | **-0.22\*\*\*** |
| Text Length | **-0.49\*\*\*** | **-0.83\*\*\*** | **-0.39\*\*\*** | **-0.43\*\*\*** | **-0.78\*\*\*** | **-0.44\*\*\*** | **-0.74\*\*\*** |
| Italics | 0.04 | 0.06 | 0.05 | 0.04 | 0.08 | 0.05 | 0.06 |
| Bold Text | −0.05 | −0.04 | 0.01 | −0.04 | −0.08 | −0.04 | −0.06 |
| Lists | 0.07 | 0.07 | 0.04 | 0.08 | 0.09 | 0.11 | 0.05 |
| Headers | 0.03 | 0.02 | 0.03 | 0.01 | −0.01 | 0.01 | 0.03 |
| Creativity/Engagement | 0.07 | 0.04 | 0.05 | 0.09 | 0.06 | 0.09 | 0.02 |
| Positive Sentiment | **-0.18\*\*\*** | **-0.31\*\*\*** | **-0.12\*** | **-0.15\*\*** | **-0.22\*\*** | **-0.18\*\*\*** | **-0.21\*\*\*** |
| Conciseness | −0.07 | −0.09 | −0.07 | −0.02 | −0.07 | −0.04 | −0.02 |
| Contrast Markers | 0.01 | 0.00 | 0.02 | −0.01 | 0.02 | −0.01 | −0.01 |
| Layout Density | −0.10 | −0.23 | **-0.15\*** | −0.11 | −0.21 | −0.13 | −0.17 |
| Causal Markers | **-0.13\*\*** | **-0.19\*\*** | −0.09 | −0.10 | −0.12 | −0.08 | −0.07 |
| Structure Counts | 0.15 | **0.35\*\*\*** | **0.16\*** | 0.11 | **0.29\*\*** | 0.12 | **0.29\*\*\*** |
| Sentiment | 0.12 | **0.24\*\*** | 0.10 | 0.11 | 0.11 | 0.09 | 0.10 |
| Readability Grade | 0.03 | 0.24 | 0.13 | −0.01 | 0.27 | 0.07 | 0.27 |
| Exclamation Density | −0.05 | −0.18 | −0.08 | −0.06 | −0.01 | 0.01 | −0.02 |
| Readability Ease | 0.10 | 0.39 | 0.18 | 0.08 | 0.45 | 0.17 | 0.43 |
| Polarity | 0.01 | −0.04 | 0.02 | 0.00 | −0.01 | 0.01 | 0.01 |
| Question Density | −0.10 | −0.09 | −0.06 | −0.11 | −0.11 | −0.09 | −0.12 |
| Paragraph Length | 0.02 | 0.04 | 0.01 | 0.04 | 0.05 | 0.03 | 0.07 |
| Code Block | 0.08 | **0.20\*\*** | 0.07 | 0.09 | **0.22\*\*\*** | **0.14\*\*** | **0.20\*\*\*** |
| Additive Markers | −0.01 | 0.05 | 0.00 | 0.00 | 0.06 | 0.00 | 0.03 |
| Summary Markers | −0.08 | −0.09 | −0.08 | −0.08 | −0.11 | −0.10 | −0.12 |
| Character Density | 0.10 | 0.25 | 0.13 | 0.12 | 0.23 | **0.20\*\*** | 0.13 |
| Exclamation Count | 0.03 | 0.13 | 0.03 | −0.01 | 0.08 | −0.03 | 0.05 |
| Lexical Ratio | −0.04 | −0.10 | 0.05 | −0.02 | −0.20 | −0.08 | −0.13 |
| Subjectivity | 0.00 | 0.02 | 0.00 | 0.02 | 0.02 | 0.01 | 0.02 |
| Sentence Length | −0.04 | −0.13 | −0.07 | −0.03 | −0.10 | −0.04 | −0.10 |
| Example Markers | 0.04 | 0.05 | 0.02 | 0.02 | 0.06 | 0.02 | 0.04 |
| Compound Sentiment | 0.11 | **0.27\*\*\*** | 0.06 | 0.08 | 0.16 | **0.13\*\*** | **0.19\*\*** |
| Question Count | **0.12\*** | 0.16 | 0.09 | **0.13\*** | 0.13 | 0.11 | 0.15 |

Significance: **\*\*\*** $p < 0.01$, **\*\*** $p < 0.05$, **\*** $p < 0.10$.

### B.6.3 Tables Chatbot Arena

Table 10: Human-LLM judgement discrepancies on Chatbot Arena (with *no* Benjamini-Yekutieli correction)

| | GPT4-turbo | GPT4.1-nano | GPT4.1 | GPT4o-mini | LLaMa-3.1-8B-It | Selene-1-Mini | Prometheus-v2 |
|---|---|---|---|---|---|---|---|
| Text Length | **-0.90***** | **-2.05***** | **-0.54***** | **-1.02***** | **-1.61***** | **-1.17***** | **-1.20***** |
| Creativity/Engagement | **-0.55***** | **-1.27***** | **-0.32***** | **-0.64***** | **-1.10***** | **-0.78***** | **-0.77***** |
| Readability Grade | 0.01 | 0.15 | −0.01 | 0.02 | 0.11 | 0.07 | 0.07 |
| Consistency | −0.17 | −0.38 | −0.12 | −0.17 | −0.32 | −0.23 | −0.23 |
| Bold Text | **0.38***** | **0.74***** | **0.25***** | **0.50***** | **0.77***** | **0.66***** | **0.62***** |
| Causal Markers | 0.09 | 0.20 | 0.08 | 0.11 | 0.15 | 0.13 | 0.11 |
| Language Quality | 0.12 | 0.23 | 0.07 | 0.15 | 0.21 | 0.15 | 0.21 |
| Example Markers | 0.11 | 0.24 | 0.08 | 0.15 | 0.20 | 0.16 | 0.15 |
| Conciseness | **-0.20*** | −0.36 | −0.08 | **-0.24*** | −0.33 | −0.27 | **-0.26*** |
| Structure Counts | 0.16 | 0.43 | 0.11 | 0.22 | 0.34 | 0.28 | 0.25 |
| Additive Markers | 0.01 | 0.08 | 0.03 | 0.04 | 0.07 | 0.03 | 0.04 |
| Polarity | 0.09 | 0.15 | 0.05 | 0.09 | 0.12 | 0.09 | 0.10 |
| Contrast Markers | −0.06 | −0.10 | −0.04 | −0.08 | −0.03 | −0.09 | −0.05 |
| Sentiment | **0.26**** | **0.57**** | **0.14*** | **0.34**** | **0.41*** | **0.35**** | **0.37**** |
| Coherence | −0.16 | **-0.47*** | −0.10 | −0.25 | −0.36 | −0.28 | −0.30 |
| Linebreak Density | 0.19 | 0.36 | 0.13 | 0.24 | 0.35 | 0.27 | 0.26 |
| Lists | −0.07 | −0.23 | −0.03 | −0.14 | −0.26 | −0.18 | −0.17 |
| Subjectivity | **0.17*** | 0.27 | 0.07 | **0.19*** | **0.28*** | **0.22*** | **0.21*** |
| Readability Ease | 0.17 | 0.49 | 0.11 | 0.22 | 0.41 | 0.28 | 0.24 |
| Paragraph Length | **0.24**** | **0.51**** | **0.17**** | **0.28**** | **0.43**** | **0.32**** | **0.30**** |
| Code Block | −0.08 | −0.20 | −0.06 | −0.09 | −0.18 | −0.14 | −0.13 |
| Question Density | 0.00 | 0.02 | −0.01 | 0.00 | 0.02 | 0.00 | 0.01 |
| List Density | 0.02 | 0.04 | −0.02 | 0.04 | 0.08 | 0.09 | 0.04 |
| Exclamation Count | 0.06 | 0.18 | 0.06 | 0.13 | 0.13 | 0.11 | 0.10 |
| Paragraph Density | **-0.23*** | **-0.50**** | **-0.16**** | **-0.26*** | **-0.46**** | **-0.36**** | **-0.37**** |
| Character Density | 0.01 | 0.00 | 0.01 | 0.01 | 0.01 | −0.02 | 0.00 |
| Count Italic | 0.24 | 0.49 | 0.19 | 0.27 | 0.45 | 0.36 | 0.37 |
| Question Count | 0.00 | 0.07 | 0.01 | 0.02 | 0.07 | 0.02 | 0.02 |
| Positive Sentiment | 0.03 | 0.12 | 0.01 | 0.08 | 0.10 | 0.07 | 0.07 |
| Compound Sentiment | 0.06 | 0.12 | 0.03 | 0.06 | 0.13 | 0.10 | 0.13 |
| Summary Markers | −0.07 | −0.08 | −0.05 | −0.06 | −0.08 | −0.06 | −0.08 |
| Lexical Ratio | **0.26*** | **0.56**** | **0.22**** | **0.30*** | **0.48*** | **0.45**** | **0.35*** |
| Relative Italic | −0.22 | −0.43 | −0.14 | −0.25 | −0.41 | −0.31 | **-0.31*** |
| Exclamation Density | **-0.18*** | −0.30 | −0.08 | −0.19 | −0.27 | −0.21 | **-0.24*** |
| Header Density | 0.00 | 0.02 | 0.01 | 0.02 | 0.00 | 0.01 | 0.00 |
| Headers | 0.02 | 0.01 | 0.03 | 0.05 | 0.01 | 0.03 | −0.01 |
| Sentence Length | −0.09 | −0.16 | −0.08 | −0.09 | −0.20 | −0.17 | −0.19 |

Significance: **\*\*\*** $p < 0.01$, **\*\*** $p < 0.05$, **\*** $p < 0.10$.

Table 11: Human-LLM judgement discrepancies on Chatbot Arena (with Benjamini-Yekutieli correction)

| | GPT4-turbo | GPT4.1-nano | GPT4.1 | GPT4o-mini | LLaMa-3.1-8B-It | Selene-1-Mini | Prometheus-v2 |
|---|---|---|---|---|---|---|---|
| Text Length | **-0.90***** | **-2.05***** | **-0.54***** | **-1.02***** | **-1.61**** | **-1.17**** | **-1.20***** |
| Creativity/Engagement | **-0.55***** | **-1.27***** | **-0.32***** | **-0.64***** | **-1.10**** | **-0.78**** | **-0.77***** |
| Readability Grade | 0.01 | 0.15 | −0.01 | 0.02 | 0.11 | 0.07 | 0.07 |
| Consistency | −0.17 | −0.38 | −0.12 | −0.17 | −0.32 | −0.23 | −0.23 |
| Bold Text | **0.38***** | **0.74**** | **0.25***** | **0.50***** | **0.77**** | **0.66***** | **0.62***** |
| Causal Markers | 0.09 | 0.20 | 0.08 | 0.11 | 0.15 | 0.13 | 0.11 |
| Language Quality | 0.12 | 0.23 | 0.07 | 0.15 | 0.21 | 0.15 | 0.21 |
| Example Markers | 0.11 | 0.24 | 0.08 | 0.15 | 0.20 | 0.16 | 0.15 |
| Conciseness | −0.20 | −0.36 | −0.08 | −0.24 | −0.33 | −0.27 | −0.26 |
| Structure Counts | 0.16 | 0.43 | 0.11 | 0.22 | 0.34 | 0.28 | 0.25 |
| Additive Markers | 0.01 | 0.08 | 0.03 | 0.04 | 0.07 | 0.03 | 0.04 |
| Polarity | 0.09 | 0.15 | 0.05 | 0.09 | 0.12 | 0.09 | 0.10 |
| Contrast Markers | −0.06 | −0.10 | −0.04 | −0.08 | −0.03 | −0.09 | −0.05 |
| Sentiment | 0.26 | 0.57 | 0.14 | 0.34 | 0.41 | 0.35 | 0.37 |
| Coherence | −0.16 | −0.47 | −0.10 | −0.25 | −0.36 | −0.28 | −0.30 |
| Linebreak Density | 0.19 | 0.36 | 0.13 | 0.24 | 0.35 | 0.27 | 0.26 |
| Lists | −0.07 | −0.23 | −0.03 | −0.14 | −0.26 | −0.18 | −0.17 |
| Subjectivity | 0.17 | 0.27 | 0.07 | 0.19 | 0.28 | 0.22 | 0.21 |
| Readability Ease | 0.17 | 0.49 | 0.11 | 0.22 | 0.41 | 0.28 | 0.24 |
| Paragraph Length | 0.24 | 0.51 | 0.17 | 0.28 | 0.43 | 0.32 | 0.30 |
| Code Block | −0.08 | −0.20 | −0.06 | −0.09 | −0.18 | −0.14 | −0.13 |
| Question Density | 0.00 | 0.02 | −0.01 | 0.00 | 0.02 | 0.00 | 0.01 |
| List Density | 0.02 | 0.04 | −0.02 | 0.04 | 0.08 | 0.09 | 0.04 |
| Exclamation Count | 0.06 | 0.18 | 0.06 | 0.13 | 0.13 | 0.11 | 0.10 |
| Paragraph Density | −0.23 | −0.50 | −0.16 | −0.26 | −0.46 | −0.36 | −0.37 |
| Character Density | 0.01 | 0.00 | 0.01 | 0.01 | 0.01 | −0.02 | 0.00 |
| Count Italic | 0.24 | 0.49 | 0.19 | 0.27 | 0.45 | 0.36 | 0.37 |
| Question Count | 0.00 | 0.07 | 0.01 | 0.02 | 0.07 | 0.02 | 0.02 |
| Positive Sentiment | 0.03 | 0.12 | 0.01 | 0.08 | 0.10 | 0.07 | 0.07 |
| Compound Sentiment | 0.06 | 0.12 | 0.03 | 0.06 | 0.13 | 0.10 | 0.13 |
| Summary Markers | −0.07 | −0.08 | −0.05 | −0.06 | −0.08 | −0.06 | −0.08 |
| Lexical Ratio | 0.26 | 0.56 | 0.22 | 0.30 | 0.48 | 0.45 | 0.35 |
| Relative Italic | −0.22 | −0.43 | −0.14 | −0.25 | −0.41 | −0.31 | −0.31 |
| Exclamation Density | −0.18 | −0.30 | −0.08 | −0.19 | −0.27 | −0.21 | −0.24 |
| Header Density | 0.00 | 0.02 | 0.01 | 0.02 | 0.00 | 0.01 | 0.00 |
| Headers | 0.02 | 0.01 | 0.03 | 0.05 | 0.01 | 0.03 | −0.01 |
| Sentence Length | −0.09 | −0.16 | −0.08 | −0.09 | −0.20 | −0.17 | −0.19 |

Significance: *** $p < 0.01$, ** $p < 0.05$, * $p < 0.10$.

### B.6.4 Tables Chatbot Arena (non-technical queries)

Table 12: Human-LLM judgement discrepancies on non-technical Chatbot Arena queries (with *no* Benjamini-Yekutieli correction)

| | GPT4-turbo | GPT4.1-nano | GPT4.1 | GPT4o-mini | LLaMa-3.1-8B-It | Selene-1-Mini | Prometheus-v2 |
|---|---|---|---|---|---|---|---|
| Text Length | **-0.90*** | **-2.18*** | **-0.55*** | **-1.09*** | **-2.45** | **-1.25*** | **-1.67*** |
| Creativity/Engagement | **-0.55*** | **-1.33*** | **-0.33*** | **-0.70*** | **-1.64** | **-0.86*** | **-1.07*** |
| Readability Grade | 0.28 | 0.78 | 0.13 | 0.35 | 0.90 | 0.46 | 0.64 |
| Consistency | **-0.24*** | **-0.56** | **-0.16*** | **-0.27*** | **-0.63*** | **-0.35*** | **-0.42*** |
| Bold Text | **0.39*** | **0.75** | **0.24*** | **0.52*** | **1.01** | **0.66*** | **0.77*** |
| Causal Markers | 0.14 | 0.32 | 0.13 | 0.19 | 0.34 | 0.21 | 0.24 |
| Language Quality | 0.24 | 0.52 | 0.14 | 0.30 | 0.62 | 0.36 | 0.47 |
| Example Markers | −0.14 | −0.27 | −0.12 | −0.16 | −0.33 | −0.21 | −0.24 |
| Conciseness | **-0.26*** | **-0.49*** | −0.11 | **-0.32*** | **-0.63*** | **-0.38*** | **-0.45*** |
| Structure Counts | 0.21 | 0.57 | 0.10 | 0.34 | 0.70 | 0.39 | 0.42 |
| Additive Markers | −0.06 | −0.06 | −0.01 | −0.04 | −0.07 | −0.08 | −0.09 |
| Polarity | 0.15 | 0.31 | 0.10 | 0.19 | 0.38 | 0.21 | 0.27 |
| Contrast Markers | −0.04 | −0.05 | −0.03 | −0.04 | 0.06 | −0.02 | 0.00 |
| Sentiment | **0.22*** | **0.46*** | 0.12 | **0.30*** | 0.44 | 0.29 | **0.38*** |
| Coherence | −0.06 | −0.23 | −0.03 | −0.10 | −0.24 | −0.13 | −0.21 |
| Linebreak Density | 0.10 | 0.21 | 0.07 | 0.18 | 0.28 | 0.19 | 0.20 |
| Lists | **-0.35*** | **-0.84*** | −0.21 | **-0.51*** | **-1.07*** | **-0.60*** | **-0.71*** |
| Subjectivity | 0.06 | 0.09 | 0.01 | 0.07 | 0.14 | 0.06 | 0.11 |
| Readability Ease | 0.17 | 0.60 | 0.11 | 0.27 | 0.65 | 0.30 | 0.39 |
| Paragraph Length | 0.09 | 0.20 | 0.07 | 0.11 | 0.24 | 0.12 | 0.13 |
| Code Block | −0.18 | −0.39 | −0.15 | −0.23 | −0.53 | −0.33 | −0.36 |
| Question Density | 0.00 | 0.06 | 0.00 | 0.02 | 0.05 | 0.01 | 0.04 |
| List Density | 0.03 | 0.02 | −0.01 | 0.03 | 0.08 | 0.08 | 0.03 |
| Exclamation Count | 0.11 | 0.31 | 0.11 | 0.22 | 0.32 | 0.20 | 0.22 |
| Paragraph Density | **-0.25*** | **-0.52*** | **-0.15*** | **-0.30*** | −0.64 | **-0.38*** | **-0.46*** |
| Character Density | 0.16 | 0.21 | 0.11 | 0.15 | 0.28 | 0.12 | 0.20 |
| Count Italic | 0.26 | 0.46 | 0.18 | 0.33 | 0.67 | 0.44 | 0.45 |
| Question Count | 0.04 | 0.15 | 0.04 | 0.08 | 0.21 | 0.07 | 0.10 |
| Positive Sentiment | −0.12 | −0.17 | −0.10 | −0.10 | −0.25 | −0.15 | −0.20 |
| Compound Sentiment | **0.22*** | **0.46*** | **0.15*** | **0.28*** | 0.59 | **0.37*** | **0.46** |
| Summary Markers | −0.05 | −0.05 | −0.05 | −0.05 | −0.09 | −0.07 | −0.09 |
| Lexical Ratio | 0.26 | **0.60*** | **0.22** | **0.36*** | 0.72 | **0.50** | 0.48 |
| Relative Italic | −0.16 | −0.30 | −0.08 | −0.18 | −0.42 | −0.25 | −0.26 |
| Exclamation Density | −0.12 | −0.20 | −0.05 | −0.13 | −0.22 | −0.13 | −0.19 |
| Header Density | 0.12 | 0.21 | 0.08 | 0.12 | 0.24 | 0.16 | 0.18 |
| Headers | −0.05 | −0.06 | 0.02 | 0.01 | −0.06 | −0.02 | −0.09 |
| Sentence Length | **-1.02** | **-1.97*** | **-0.59*** | **-1.13*** | **-2.55*** | **-1.47*** | **-1.87** |

Significance: *** $p < 0.01$, ** $p < 0.05$, * $p < 0.10$.

Table 13: Human-LLM judgement discrepancies on non-technical Chatbot Arena queries (with Benjamini-Yekutieli correction)

| | GPT4-turbo | GPT4.1-nano | GPT4.1 | GPT4o-mini | LLaMa-3.1-8B-It | Selene-1-Mini | Prometheus-v2 |
|---|---|---|---|---|---|---|---|
| Text Length | **-0.90**\*\* | **-2.18**\* | **-0.55**\*\* | **-1.09**\* | −2.45 | −1.25 | −1.67 |
| Creativity/Engagement | **-0.55**\*\* | **-1.33**\* | **-0.33**\*\* | **-0.70**\* | −1.64 | −0.86 | −1.07 |
| Readability Grade | 0.28 | 0.78 | 0.13 | 0.35 | 0.90 | 0.46 | 0.64 |
| Consistency | −0.24 | −0.56 | −0.16 | −0.27 | −0.63 | −0.35 | −0.42 |
| Bold Text | 0.39 | 0.75 | 0.24 | **0.52**\* | 1.01 | 0.66 | 0.77 |
| Causal Markers | 0.14 | 0.32 | 0.13 | 0.19 | 0.34 | 0.21 | 0.24 |
| Language Quality | 0.24 | 0.52 | 0.14 | 0.30 | 0.62 | 0.36 | 0.47 |
| Example Markers | −0.14 | −0.27 | −0.12 | −0.16 | −0.33 | −0.21 | −0.24 |
| Conciseness | −0.26 | −0.49 | −0.11 | −0.32 | −0.63 | −0.38 | −0.45 |
| Structure Counts | 0.21 | 0.57 | 0.10 | 0.34 | 0.70 | 0.39 | 0.42 |
| Additive Markers | −0.06 | −0.06 | −0.01 | −0.04 | −0.07 | −0.08 | −0.09 |
| Polarity | 0.15 | 0.31 | 0.10 | 0.19 | 0.38 | 0.21 | 0.27 |
| Contrast Markers | −0.04 | −0.05 | −0.03 | −0.04 | 0.06 | −0.02 | 0.00 |
| Sentiment | 0.22 | 0.46 | 0.12 | 0.30 | 0.44 | 0.29 | 0.38 |
| Coherence | −0.06 | −0.23 | −0.03 | −0.10 | −0.24 | −0.13 | −0.21 |
| Linebreak Density | 0.10 | 0.21 | 0.07 | 0.18 | 0.28 | 0.19 | 0.20 |
| Lists | −0.35 | −0.84 | −0.21 | −0.51 | −1.07 | −0.60 | −0.71 |
| Subjectivity | 0.06 | 0.09 | 0.01 | 0.07 | 0.14 | 0.06 | 0.11 |
| Readability Ease | 0.17 | 0.60 | 0.11 | 0.27 | 0.65 | 0.30 | 0.39 |
| Paragraph Length | 0.09 | 0.20 | 0.07 | 0.11 | 0.24 | 0.12 | 0.13 |
| Code Block | −0.18 | −0.39 | −0.15 | −0.23 | −0.53 | −0.33 | −0.36 |
| Question Density | 0.00 | 0.06 | 0.00 | 0.02 | 0.05 | 0.01 | 0.04 |
| List Density | 0.03 | 0.02 | −0.01 | 0.03 | 0.08 | 0.08 | 0.03 |
| Exclamation Count | 0.11 | 0.31 | 0.11 | 0.22 | 0.32 | 0.20 | 0.22 |
| Paragraph Density | −0.25 | −0.52 | −0.15 | −0.30 | −0.64 | −0.38 | −0.46 |
| Character Density | 0.16 | 0.21 | 0.11 | 0.15 | 0.28 | 0.12 | 0.20 |
| Count Italic | 0.26 | 0.46 | 0.18 | 0.33 | 0.67 | 0.44 | 0.45 |
| Question Count | 0.04 | 0.15 | 0.04 | 0.08 | 0.21 | 0.07 | 0.10 |
| Positive Sentiment | −0.12 | −0.17 | −0.10 | −0.10 | −0.25 | −0.15 | −0.20 |
| Compound Sentiment | 0.22 | 0.46 | 0.15 | 0.28 | 0.59 | 0.37 | 0.46 |
| Summary Markers | −0.05 | −0.05 | −0.05 | −0.05 | −0.09 | −0.07 | −0.09 |
| Lexical Ratio | 0.26 | 0.60 | 0.22 | 0.36 | 0.72 | 0.50 | 0.48 |
| Relative Italic | −0.16 | −0.30 | −0.08 | −0.18 | −0.42 | −0.25 | −0.26 |
| Exclamation Density | −0.12 | −0.20 | −0.05 | −0.13 | −0.22 | −0.13 | −0.19 |
| Header Density | 0.12 | 0.21 | 0.08 | 0.12 | 0.24 | 0.16 | 0.18 |
| Headers | −0.05 | −0.06 | 0.02 | 0.01 | −0.06 | −0.02 | −0.09 |
| Sentence Length | −1.02 | −1.97 | −0.59 | −1.13 | −2.55 | −1.47 | −1.87 |

Significance: **\*\*\*** $p < 0.01$, **\*\*** $p < 0.05$, **\*** $p < 0.10$.

### B.6.5 Tables Chatbot Arena (technical queries)

Table 14: Human-LLM judgement discrepancies on technical Chatbot Arena queries (with *no* Benjamini-Yekutieli correction)

| | GPT4-turbo | GPT4.1-nano | GPT4.1 | GPT4o-mini | LLaMa-3.1-8B-It | Selene-1-Mini | Prometheus-v2 |
|---|---|---|---|---|---|---|---|
| Text Length | **-1.06*** | **-2.36** | **-0.60*** | **-1.03*** | **-1.10** | **-1.07** | **-0.92*** |
| Creativity/Engagement | **-0.60*** | **-1.32*** | −0.27 | **-0.52*** | **-0.62*** | **-0.58*** | **-0.48*** |
| Readability Grade | 0.27 | 0.58 | 0.17 | 0.23 | 0.32 | 0.31 | 0.25 |
| Consistency | 0.08 | 0.18 | 0.04 | 0.12 | 0.10 | 0.13 | 0.07 |
| Bold Text | **0.45** | **0.90** | **0.28** | **0.52*** | **0.62*** | **0.65*** | **0.52*** |
| Causal Markers | 0.04 | 0.12 | 0.04 | 0.05 | 0.05 | 0.07 | 0.03 |
| Language Quality | 0.12 | 0.17 | 0.05 | 0.17 | 0.12 | 0.06 | 0.16 |
| Example Markers | **0.30*** | **0.64*** | **0.21** | **0.33** | **0.35*** | **0.35** | **0.28** |
| Conciseness | −0.05 | −0.16 | 0.00 | −0.10 | −0.05 | −0.03 | −0.04 |
| Structure Counts | 0.15 | 0.41 | 0.12 | 0.19 | 0.14 | 0.20 | 0.15 |
| Additive Markers | 0.06 | 0.16 | 0.05 | 0.07 | 0.09 | 0.08 | 0.08 |
| Polarity | −0.03 | −0.11 | −0.05 | −0.06 | −0.09 | −0.09 | −0.06 |
| Contrast Markers | −0.12 | −0.22 | −0.08 | −0.16 | −0.13 | −0.20 | −0.12 |
| Sentiment | 0.40 | **0.95*** | 0.22 | **0.45*** | 0.44 | 0.46 | 0.37 |
| Coherence | **-1.00** | **-2.24** | **-0.55** | **-1.10** | **-1.19** | **-1.15** | **-0.94** |
| Linebreak Density | 0.33 | 0.67 | 0.22 | 0.31 | 0.43 | 0.36 | 0.28 |
| Lists | 0.09 | 0.07 | 0.08 | 0.03 | −0.01 | 0.02 | 0.03 |
| Subjectivity | **0.40** | 0.70 | **0.21*** | **0.40** | **0.46** | **0.48** | **0.34** |
| Readability Ease | 0.42 | 0.96 | 0.26 | 0.39 | 0.48 | 0.46 | 0.33 |
| Paragraph Length | **0.54** | **1.17** | **0.35** | **0.58** | **0.59** | **0.58** | **0.45** |
| Code Block | −0.07 | −0.19 | −0.05 | −0.07 | −0.11 | −0.11 | −0.08 |
| Question Density | −0.20 | −0.45 | −0.14 | −0.19 | −0.13 | −0.14 | −0.15 |
| List Density | 0.12 | 0.30 | 0.05 | 0.20 | 0.21 | 0.25 | 0.15 |
| Exclamation Count | −0.01 | 0.00 | −0.02 | 0.03 | 0.02 | 0.01 | 0.05 |
| Paragraph Density | −0.15 | −0.42 | −0.13 | −0.15 | −0.25 | −0.26 | −0.22 |
| Character Density | 0.02 | 0.09 | 0.02 | 0.02 | 0.04 | 0.02 | 0.02 |
| Count Italic | −0.05 | −0.01 | 0.02 | −0.07 | −0.09 | −0.08 | 0.01 |
| Question Count | 0.04 | 0.13 | 0.02 | 0.02 | 0.06 | 0.05 | 0.05 |
| Positive Sentiment | **0.59*** | **1.23*** | **0.37*** | **0.59*** | **0.73** | **0.70*** | **0.60** |
| Compound Sentiment | −0.27 | −0.54 | −0.20 | −0.29 | −0.31 | −0.34 | −0.24 |
| Summary Markers | −0.09 | −0.11 | −0.05 | −0.06 | −0.07 | −0.05 | −0.07 |
| Lexical Ratio | −0.07 | −0.12 | 0.01 | −0.11 | −0.13 | −0.04 | −0.10 |
| Relative Italic | −0.21 | −0.46 | −0.15 | −0.23 | −0.25 | −0.23 | −0.23 |
| Exclamation Density | −1.36 | −2.18 | −0.81 | −1.43 | −1.60 | −1.53 | −1.41 |
| Header Density | −0.04 | −0.05 | 0.00 | 0.01 | −0.03 | −0.05 | −0.02 |
| Headers | 0.06 | 0.07 | 0.03 | 0.07 | 0.05 | 0.07 | 0.03 |
| Sentence Length | 0.11 | 0.19 | 0.02 | 0.13 | 0.13 | 0.10 | 0.04 |

Significance: *** $p < 0.01$, ** $p < 0.05$, * $p < 0.10$.

Table 15: Human-LLM judgement discrepancies on technical Chatbot Arena queries (with Benjamini-Yekutieli correction)

| | GPT4-turbo | GPT4.1-nano | GPT4.1 | GPT4o-mini | LLaMa-3.1-8B-It | Selene-1-Mini | Prometheus-v2 |
|---|---|---|---|---|---|---|---|
| Text Length | −1.06 | −2.36 | −0.60 | −1.03 | −1.10 | −1.07 | −0.92 |
| Creativity/Engagement | −0.60 | −1.32 | −0.27 | −0.52 | −0.62 | −0.58 | −0.48 |
| Readability Grade | 0.27 | 0.58 | 0.17 | 0.23 | 0.32 | 0.31 | 0.25 |
| Consistency | 0.08 | 0.18 | 0.04 | 0.12 | 0.10 | 0.13 | 0.07 |
| Bold Text | 0.45 | 0.90 | 0.28 | 0.52 | 0.62 | 0.65 | 0.52 |
| Causal Markers | 0.04 | 0.12 | 0.04 | 0.05 | 0.05 | 0.07 | 0.03 |
| Language Quality | 0.12 | 0.17 | 0.05 | 0.17 | 0.12 | 0.06 | 0.16 |
| Example Markers | 0.30 | 0.64 | 0.21 | 0.33 | 0.35 | 0.35 | 0.28 |
| Conciseness | −0.05 | −0.16 | 0.00 | −0.10 | −0.05 | −0.03 | −0.04 |
| Structure Counts | 0.15 | 0.41 | 0.12 | 0.19 | 0.14 | 0.20 | 0.15 |
| Additive Markers | 0.06 | 0.16 | 0.05 | 0.07 | 0.09 | 0.08 | 0.08 |
| Polarity | −0.03 | −0.11 | −0.05 | −0.06 | −0.09 | −0.09 | −0.06 |
| Contrast Markers | −0.12 | −0.22 | −0.08 | −0.16 | −0.13 | −0.20 | −0.12 |
| Sentiment | 0.40 | 0.95 | 0.22 | 0.45 | 0.44 | 0.46 | 0.37 |
| Coherence | −1.00 | −2.24 | −0.55 | −1.10 | −1.19 | −1.15 | −0.94 |
| Linebreak Density | 0.33 | 0.67 | 0.22 | 0.31 | 0.43 | 0.36 | 0.28 |
| Lists | 0.09 | 0.07 | 0.08 | 0.03 | −0.01 | 0.02 | 0.03 |
| Subjectivity | 0.40 | 0.70 | 0.21 | 0.40 | 0.46 | 0.48 | 0.34 |
| Readability Ease | 0.42 | 0.96 | 0.26 | 0.39 | 0.48 | 0.46 | 0.33 |
| Paragraph Length | 0.54 | 1.17 | 0.35 | 0.58 | 0.59 | 0.58 | 0.45 |
| Code Block | −0.07 | −0.19 | −0.05 | −0.07 | −0.11 | −0.11 | −0.08 |
| Question Density | −0.20 | −0.45 | −0.14 | −0.19 | −0.13 | −0.14 | −0.15 |
| List Density | 0.12 | 0.30 | 0.05 | 0.20 | 0.21 | 0.25 | 0.15 |
| Exclamation Count | −0.01 | 0.00 | −0.02 | 0.03 | 0.02 | 0.01 | 0.05 |
| Paragraph Density | −0.15 | −0.42 | −0.13 | −0.15 | −0.25 | −0.26 | −0.22 |
| Character Density | 0.02 | 0.09 | 0.02 | 0.02 | 0.04 | 0.02 | 0.02 |
| Count Italic | −0.05 | −0.01 | 0.02 | −0.07 | −0.09 | −0.08 | 0.01 |
| Question Count | 0.04 | 0.13 | 0.02 | 0.02 | 0.06 | 0.05 | 0.05 |
| Positive Sentiment | 0.59 | 1.23 | 0.37 | 0.59 | 0.73 | 0.70 | 0.60 |
| Compound Sentiment | −0.27 | −0.54 | −0.20 | −0.29 | −0.31 | −0.34 | −0.24 |
| Summary Markers | −0.09 | −0.11 | −0.05 | −0.06 | −0.07 | −0.05 | −0.07 |
| Lexical Ratio | −0.07 | −0.12 | 0.01 | −0.11 | −0.13 | −0.04 | −0.10 |
| Relative Italic | −0.21 | −0.46 | −0.15 | −0.23 | −0.25 | −0.23 | −0.23 |
| Exclamation Density | −1.36 | −2.18 | −0.81 | −1.43 | −1.60 | −1.53 | −1.41 |
| Header Density | −0.04 | −0.05 | 0.00 | 0.01 | −0.03 | −0.05 | −0.02 |
| Headers | 0.06 | 0.07 | 0.03 | 0.07 | 0.05 | 0.07 | 0.03 |
| Sentence Length | 0.11 | 0.19 | 0.02 | 0.13 | 0.13 | 0.10 | 0.04 |

Significance: *** $p < 0.01$, ** $p < 0.05$, * $p < 0.10$.

## C  Additional theoretical results and proofs

### C.1  Conditions

The following two conditions are required for Proposition 3.1.

**Condition C.1.** *The true parameter $(\eta^*, Z_{1:n}^{l,*})$ lies in the interior of a compact subset of $\Theta_\eta \times \mathcal{Z}^n \subset \mathbb{R}^{K+n}$, and is the unique minimizer of the population loss $Q_n(\eta, z_{1:n})$.*

Denote

$$A_n := \sqrt{\frac{2}{\pi}} \sum_{i=1}^{n} \sum_{k=0}^{K} \frac{\nabla_{(\eta, z_{1:n})} p_k(\eta^*, Z_i^{l,*}) \nabla_{(\eta, z_{1:n})} p_k(\eta^*, Z_i^{l,*})^\top}{\sqrt{p_{ik}(1 - p_{ik})}},$$

where $p_{ik} = p_k(\eta^*, Z_i^{l,*})$. Let $\xi_{ik} \in \{-1, 1\}$, $i = 1, \ldots, n$, $k = 0, \ldots, K$ follow i.i.d. Rademacher(1/2) distribution. Define a vector $S_n := \sum_{i=1}^{n} \sum_{k=0}^{K} \xi_{ik} \nabla_{(\eta, z_{1:n})} p_k(\eta^*, Z_i^{l,*})$, and its variance $B_n = \mathrm{Var}(S_n)$. Denote $\Sigma = A_n^{-1} B_n A_n^{-1}$.

**Condition C.2.** *Let matrix $A_n$ be positive definite, and the variance $B_n = \mathrm{Var}(S_n)$ be finite.*

Below we state the conditions of Theorem 3.2.

**Condition C.3.** *The observations $\{(Y_i^h, X_i, I_i, O_i)\}_{i=1}^{n}$ are i.i.d. and the ordinallogit model*

$$\mathbb{P}(Y_i^h = k \mid I_i, O_i) = l_k(\theta; Z_i^l, X_i) = p_k(\alpha_1, \ldots, \alpha_K, (1/\beta)Z_i^l - (1/\beta)\gamma^T X_i)$$

*holds for some true cut-points $\alpha_1^* < \cdots < \alpha_K^*$, coefficients $\beta^*$ and $\gamma^*$, and latent scores $Z_i^{l,*}$.*

**Condition C.4.** *The true parameter vector $\theta^* = (\alpha_1^*, \ldots, \alpha_K^*, \beta^*, \gamma^*)$ lies in the interior of a parameter space $\Theta$, and $\theta \mapsto \mathbb{E}[\nabla_\theta \log l_{Y_i^h}(\theta; Z_i^l, X_i)]$ has a unique root at $\theta^*$.*

**Condition C.5.** *Within a neighborhood of $\theta^*$, the expectations $\mathbb{E}[\|\nabla_\theta \log l_{Y_i^h}(\theta; Z_i^l, X_i)\|^2]$ and $\mathbb{E}[\|\nabla_\theta^2 \log l_{Y_i^h}(\theta; Z_i^l, X_i)\|]$ are finite. Moreover, the Fisher information matrix $\mathcal{I}(\theta^*) = -\mathbb{E}[\nabla_\theta^2 \log l_{Y_i^h}(\theta^*; Z_i^{l,*}, X_i)]$ is positive definite, and the expected mixed derivative $G(\theta^*) = \mathbb{E}[\nabla_{(\eta, Z_i^l)} \nabla_\theta \log l_{Y_i^h}(\theta^*; Z_i^{l,*}, X_i)]$ exists and is finite.*

### C.2  Extended Theorem 3.2

We present an extended version of Theorem 3.2, addressing a more general case where the CoT sample size $m_n$ used to estimate $\mathbb{P}(Y_i^h = k \mid I_i, O_i)$, $i = 1, \ldots, n$ grows proportionally with $n$.

**Theorem C.6** (Asymptotic normality of $(\hat{\beta}, \hat{\gamma})$)**.** *Under Conditions C.3–C.5, form the MLE $\hat{\theta}_n = (\hat{\alpha}_{1,n}, \ldots, \hat{\alpha}_{K,n}, \hat{\beta}_n, \hat{\gamma}_n)$ by maximizing the loglikelihood*

$$\ell_n(\theta; \hat{Z}^l, X) = \sum_{i=1}^{n} \sum_{k=0}^{K} \mathbf{1}\{Y_i^h = k\} \log l_k(\theta; \hat{Z}_i^l, X_i).$$

*w.r.t. $\theta = (\alpha_1, \ldots, \alpha_K, \beta, \gamma)$.*

*(a) If $\eta_1, \ldots, \eta_K$ and $\{Z_i^l\}_{i=1}^{n}$ were known, then*

$$\sqrt{n}(\hat{\theta}_n - \theta_0) \xrightarrow{d} \mathcal{N}(0, \mathcal{I}(\theta^*)^{-1}), \qquad \sqrt{n}\begin{pmatrix} \hat{\beta}_n - \beta_0 \\ \hat{\gamma}_n - \gamma_0 \end{pmatrix} \xrightarrow{d} \mathcal{N}(0, \{\mathcal{I}(\theta^*)^{-1}\}_{(\beta, \gamma)}).$$

*(b) If the CoT prompting strategy is used to estimate $\mathbb{P}(Y_i^l = k \mid I_i, O_i)$, also assume Conditions C.1 and C.2, and let $n/m_n \to c \in [0, \infty)$ as $n \to \infty$. Then*

$$\sqrt{n}(\hat{\theta}_n - \theta_0) \xrightarrow{d} \mathcal{N}(0, \mathcal{I}(\theta^*)^{-1} U \mathcal{I}(\theta^*)^{-1}), \quad U := \mathrm{Var}[s_i(\theta_0)] + c\, G\Sigma G^\top.$$

*Consequently,*

$$\sqrt{n}\begin{pmatrix} \hat{\beta}_n - \beta_0 \\ \hat{\gamma}_n - \gamma_0 \end{pmatrix} \xrightarrow{d} \mathcal{N}\left(0, \{\mathcal{I}(\theta^*)^{-1} U \mathcal{I}(\theta^*)^{-1}\}_{(\beta, \gamma)}\right).$$

*When $c = 0$ (that is, $m_n \gg n$), the extra variance term drops out and the estimators attain the efficiency bound of part (a). For any fixed $c > 0$ the variance is inflated by the second term, quantifying the price of estimating $(\eta, Z^l)$.*

## C.3 Inference for a differentiable function of the parameter

Consider a differentiable function $m(\theta)$. Under previous conditions, the delta-method yields

$$\sqrt{n}\big(\hat{m} - m(\theta^*)\big) \xrightarrow{d} \mathcal{N}\Big(0, \nabla_\theta m(\theta^*)^\top \mathcal{I}(\theta^*)^{-1} \nabla_\theta m(\theta^*)\Big).$$

Set $\hat{\sigma}^2 = \frac{1}{n}\nabla_\theta m(\hat{\theta}_n)^\top \hat{V} \nabla_\theta m(\hat{\theta}_n)$. An approximate $100(1-\alpha)\%$ confidence interval (CI) for $m(\theta^*)$ is $\big[\hat{m} \pm z_{1-\alpha/2}\hat{\sigma}\big]$, where $z_{1-\alpha/2}$ is the standard normal quantile. Under previous regularity conditions, the coverage probability of this CI converges to $1-\alpha$ as $n$ grows.

**Prediction interval.** This result can be used to build a CI for the prediction of a new observation $(I_{\text{new}}, O_{\text{new}}, X_{\text{new}}, \hat{Z}^l_{\text{new}})$, defined as $\hat{m} = \sum_{k=0}^K k\hat{p}_k$, where $\hat{p}_k = p_k(\hat{\alpha}_1, \ldots, \hat{\alpha}_K, z_{\text{new}})$, and $z_{\text{new}} = \hat{\beta}^{-1}\hat{Z}^l_{\text{new}} - \hat{\beta}^{-1}\hat{\gamma}^\top X_{\text{new}}$. This corresponds to the specific function $m(\theta) = \sum_{k=0}^K k\, p_k(\alpha_1, \ldots, \alpha_K, z(\theta))$, with $z(\theta) = \beta^{-1}Z^l_{\text{new}} - \beta^{-1}\gamma^\top X_{\text{new}}$.

**Partial effect of a covariate.** We also consider the "partial effect" of covariate $X_j$ on the probability of class $k$:

$$PE_{k,j} = \frac{\partial}{\partial X_j} p_k\big(\hat{\alpha},\ \hat{\beta}^{-1}Z^l - \hat{\beta}^{-1}\hat{\gamma}^\top X\big).$$

It is the local, ceteris paribus change in the models predicted probability of class $k$ when covariate $X_j$ is nudged by one unit, holding the latent index $Z_l$ constant. In other words, it answers the question: All else equal, how much does the LLM-judges probability of assigning class $k$ change when feature $X_j$ is perturbed by one unit? At $\hat{\theta} = (\hat{\alpha}, \hat{\beta}, \hat{\gamma})$, $PE_{k,j} = p'_k(\hat{\alpha}, \hat{z})(-\hat{\gamma}_j/\hat{\beta})$, where $p'_k(\alpha, z) = -\sigma'(\alpha_{k+1} - z) + \sigma'(\alpha_k - z)$, $\sigma'(t) = \sigma(t)(1 - \sigma(t))$, with $\hat{z} = z(Z^l_{new}, X_{new}; \hat{\beta}, \hat{\gamma})$. Define the scalar mapping $m(\theta) = PE_{k,j}(\theta)$. Then $\sqrt{n}\big(m(\hat{\theta}) - m(\theta^*)\big) \xrightarrow{d} \mathcal{N}\big(0, \nabla m(\theta^*)^\top V \nabla m(\theta^*)\big)$, where the gradient $\nabla m$ is taken w.r.t. $\theta = (\alpha, \beta, \gamma)$. An approximate $1-\alpha$ confidence interval for $PE_{k,j}$ is

$$PE_{k,j}(\hat{\theta}) \pm z_{1-\alpha/2}\sqrt{\frac{1}{n}\nabla m(\hat{\theta})^\top \hat{V} \nabla m(\hat{\theta})},$$

where $\hat{V}$ estimates the asymptotic covariance of $\hat{\theta}$ and $z_{1-\alpha/2}$ is the standard normal quantile.

## C.4 Proofs

### C.4.1 Proposition 3.1

*Proof of Proposition 3.1.* Define, for each $i$,

$$g_i(\eta, z_{1:n}) := \sum_{k=0}^K \text{sgn}\big(p_k(\eta, z_i) - \hat{p}_{ik,m_n}\big) \nabla_{(\eta, z_{1:n})} p_k(\eta, z_i),$$

so that $G_{n,m_n}(\eta, z_{1:n}) := \sum_{i=1}^n g_i(\eta, z_{1:n})$ is a measurable sub-gradient of $Q_{n,m_n}$. First-order optimality of $(\hat{\eta}, \hat{Z}^l_{1:n})$ gives $0 \in G_{n,m_n}(\hat{\eta}, \hat{Z}^l_{1:n})$. Write

$$r_{ik}(\eta, z_{1:n}) := p_k(\eta, z_i) - \hat{p}_{ik,m_n}, \qquad \varepsilon_{ik} := \hat{p}_{ik,m_n} - p_{ik}.$$

By the model assumptions, $r_{ik}(\eta^*, Z^{l,*}_{1:n}) = -\varepsilon_{ik}$. Because $p_k$ is continuous and differentiable, and $p_{ik} = p_k(\eta^*, Z^{l,*}_{1:n}) \in (0,1)$ which rules out kinks of the absolute value,

$\text{sgn}(r_{ik}(\eta, z_{1:n}))$

$$= \text{sgn}(-\varepsilon_{ik}) + 2f_{ik}(0) \nabla_{(\eta, z_{1:n})} p_k(\eta^*, z_i^*)^\top \begin{pmatrix} \eta - \eta^* \\ z_{1:n} - Z^{l,*}_{1:n} \end{pmatrix} + o_p\left(\left\|\begin{pmatrix} \eta - \eta^* \\ z_{1:n} - Z^{l,*}_{1:n} \end{pmatrix}\right\|\right),$$

where $f_{ik}(0) = \sqrt{m_n}\big[2\pi p_{ik}(1-p_{ik})\big]^{-1/2}$ is the asymptotic density of $\sqrt{m_n}\varepsilon_{ik}$ at 0. Setting $(\eta, z_{1:n}) = (\hat{\eta}, \hat{Z}^l_{1:n})$ and summing over $k$ and $i$,

$$0 = S_{n,m_n} + A_{n,m_n} \begin{pmatrix} \hat{\eta} - \eta^* \\ \hat{Z}^l_{1:n} - Z^{l,*}_{1:n} \end{pmatrix} + r_{n,m_n},$$

where

$$S_{n,m_n} := \sum_{i=1}^{n} \sum_{k=0}^{K} \operatorname{sgn}(-\varepsilon_{ik}) \, \nabla_{(\eta, z_{1:n})} p_k(\eta^*, z_i^*),$$

$$A_{n,m_n} := 2 \sum_{i,k} f_{ik}(0) \, \nabla_{(\eta, z_{1:n})} p_k(\eta^*, Z_i^{l,*}) \nabla_{(\eta, z_{1:n})} p_k(\eta^*, Z_i^{l,*})^{\top},$$

and

$$r_{n,m_n} = o_p\left( \left\| \begin{pmatrix} \hat{\eta} - \eta^* \\ \hat{Z}_{1:n}^l - Z_{1:n}^{l,*} \end{pmatrix} \right\| \right).$$

Since $\operatorname{sgn}(-\varepsilon_{ik}) \in \{-1, 1\}$ and $n(K+1)$ is fixed, $S_{n,m_n} = O_p(1)$. For every $i$ and $k$, we have $p_{ik} \in (0, 1)$, so $A_{n,m_n} = \sqrt{m_n} A_n + o(\sqrt{m_n})$ with $A_n$ in the statement. Rearranging the expansion,

$$\sqrt{m_n} \begin{pmatrix} \hat{\eta} - \eta^* \\ \hat{Z}_{1:n}^l - Z_{1:n}^{l,*} \end{pmatrix} = -A_n^{-1} S_{n,m_n} + o_p(1).$$

Because $\sqrt{m_n} \varepsilon_{ik} \xrightarrow{d} N\big(0, p_{ik}(1 - p_{ik})\big)$, the continuous-mapping theorem gives $\operatorname{sgn}(-\varepsilon_{ik}) \xrightarrow{d} \xi_{ik}$ with $\xi_{ik} \sim \operatorname{Rad}(1/2)$, the Rademacher distribution, and independence across $(i, k)$. Therefore $S_{n,m_n} \xrightarrow{d} S_n := \sum_{i=1}^{n} \sum_{k=0}^{K} \xi_{ik} \nabla_{(\eta, z_{1:n})} p_k(\eta^*, z_i^*)$, $\xi_{ik} \overset{\text{i.i.d.}}{\sim} \operatorname{Rad}(1/2)$ and the continuous mapping theorem yields the desired convergence in distribution to $-A_n^{-1} S_n$. Therefore, the asymptotic distribution of $\sqrt{m_n}\big[(\hat{\eta}, \hat{Z}_{1:n}^l) - (\eta^*, Z_{1:n}^{l,*})\big]$ has mean 0 and variance $\Sigma = A_n^{-1} \operatorname{Var}(S_n) A_n^{-1}$. $\quad \square$

### C.4.2 Theorem 3.2

Since Theorem 3.2 can be derived from Theorem C.6, we present the proof of the latter.

*Proof of Theorem C.6.* The proof basically follows the M-estimation consistency framework [42, chapter 5]. Denote

$$\ell_{n,i}(\theta; Z_i^l, X_i) = \sum_{k=0}^{K} \mathbf{1}\{Y_i^h = k\} \log l_k(\theta; Z_i^l, X_i), \quad s_i(\theta, Z) = \nabla_\theta \ell_{n,i}(\theta; Z, X_i).$$

Let $S_n(\theta) := n^{-1} \sum_{i=1}^{n} s_i(\theta, Z_i^l)$, $\hat{S}_n(\theta) := n^{-1} \sum_{i=1}^{n} s_i(\theta, \hat{Z}_i^l)$, and similarly define the (negative) Hessians:

$$H_n(\theta) = -\frac{1}{n} \sum_{i=1}^{n} \nabla_\theta^2 \ell_{n,i}(\theta; Z_i^l, X_i), \quad \hat{H}_n(\theta) = -\frac{1}{n} \sum_{i=1}^{n} \nabla_\theta^2 \ell_{n,i}(\theta; \hat{Z}_i^l, X_i).$$

When the true values of $\{Z_i^l\}_{i=1}^{n}$ are known (when the log-probabilities are used), by Assumptions C.3–C.5 and a uniform law of large numbers (ULLN), $S_n(\theta) \to \mathbb{E}[s_i(\theta)]$ uniformly on $\Theta$. Since the limit has a unique zero at $\theta^*$ (Condition C.4), the arg-max theorem yields $\hat{\theta}_n \xrightarrow{p} \theta^*$.

Using a mean-value expansion around $\theta_0$,

$$0 = \sqrt{n}\, S_n(\hat{\theta}_n) = \sqrt{n}\, S_n(\theta^*) - H_n(\bar{\theta}_n) \sqrt{n}(\hat{\theta}_n - \theta^*),$$

where $\bar{\theta}_n$ lies on the segment $[\theta^*, \hat{\theta}_n]$. By Conditions C.4–C.5, $H_n(\bar{\theta}_n) \xrightarrow{p} \mathcal{I}(\theta^*)$. By CLT for the score, $\sqrt{n}\, S_n(\theta_0) \xrightarrow{d} \mathcal{N}(0, \operatorname{Var}[s_i(\theta_0)]) = \mathcal{N}(0, \mathcal{I})$. Slutskys theorem gives

$$\sqrt{n}(\hat{\theta}_n - \theta_0) = H_n(\bar{\theta}_n)^{-1} \sqrt{n}\, S_n(\theta_0) \xrightarrow{d} \mathcal{I}(\theta^*)^{-1/2} \mathcal{N}(0, I) = \mathcal{N}(0, \mathcal{I}(\theta^*)^{-1}),$$

establishing part (a).

When latent scores $\{Z_i^l\}_{i=1}^{n}$ are estimated by the CoT prompting strategy, by Conditions C.4 and C.5, $\ell_i(\theta; Z^l, X)$ is jointly continuous in $(\theta, Z^l)$ and bounded by an integrable envelope. By Proposition 3.1, $\sup_{1 \le i \le n} \|\hat{Z}_i^l - Z_i^l\| = O_p\big(m_n^{-1/2}\big) = o_p(1)$, and we have

$$\sup_{\theta \in \Theta} \left| \frac{1}{n} \sum_{i=1}^{n} \ell_{n,i}(\theta; \hat{Z}_i^l, X_i) - \frac{1}{n} \sum_{i=1}^{n} \ell_{n,i}(\theta; Z_i^l, X_i) \right| = o_p(1).$$

Combining this with the uniform law of large numbers for the known regressors $(Z_i^l, X_i)$,

$$\sup_{\theta \in \Theta} \left| \frac{1}{n} \sum_{i=1}^{n} \ell_{n,i}(\theta; Z_i^l, X_i) - \mathbb{E}[\ell_{n,i}(\theta; Z_i^l, X_i)] \right| \xrightarrow{p} 0,$$

we obtain $\sup_{\theta \in \Theta} |n^{-1} \sum_{i=1}^{n} \ell_{n,i}(\theta; \hat{Z}_i^l, X_i) - \mathbb{E}[\ell_{n,i}(\theta; Z_i^l, X_i)]| \xrightarrow{p} 0$. Hence

$$\hat{\theta}_n \xrightarrow{p} \theta^* \tag{C.1}$$

by the arg-max theorem and uniqueness of $\theta^*$. For every $i$, by the mean-value theorem in variable $Z^l$,

$$s_i(\theta^*, \hat{Z}_i^l) - s_i(\theta^*, Z_i^l) = \nabla_Z s_i(\theta^*, \tilde{Z}_i^l)^\top (\hat{Z}_i^l - Z_i^l)$$

for some $\tilde{Z}_i^l$ on the segment $[Z_i^l, \hat{Z}_i^l]$. Summing over $i$ and dividing by $n$ gives

$$\hat{S}_n(\theta^*) - S_n(\theta^*) = \frac{1}{n} \sum_{i=1}^{n} \nabla_Z s_i(\theta^*, \tilde{Z}_i^l)^\top (\hat{Z}_i^l - Z_i^l). \tag{C.2}$$

By Condition C.4, $\sup_w \|\nabla_Z s_i(\theta^*, Z)\| \le C_i$ for some integrable $C_i$ (dominated convergence applies). Since $\|\tilde{Z}_i^l - Z_i^l\| \le \|\hat{Z}_i^l - Z_i^l\| = O_p(m_n^{-1/2})$,

$$\frac{1}{n} \sum_{i=1}^{n} \|\nabla_Z s_i(\theta^*, \tilde{Z}_i^l) - \nabla_Z s_i(\theta^*, Z_i^l)\| = O_p(m_n^{-1/2}) = o_p(n^{-1/2}),$$

because $n/m_n \to c < \infty$. Hence one can replace $\nabla_w s_i(\theta^*, \tilde{Z}_i^l)$ by $\nabla_w s_i(\theta^*, Z_i^l)$ in (C.2) at the cost of an $o_p(n^{-1/2})$ term. Define therefore

$$\Delta_n := \frac{1}{n} \sum_{i=1}^{n} \nabla_Z s_i(\theta^*, Z_i^l)^\top (\hat{Z}_i^l - Z_i^l), \quad \text{so that} \ \ \hat{S}_n(\theta^*) = S_n(\theta^*) + \Delta_n + o_p(n^{-1/2}).$$

By Proposition 3.1, there exist i.i.d. random vectors $e_{i,m}$ with mean 0 and variance $I$ such that

$$\hat{Z}_i^l - Z_i^l = \frac{1}{\sqrt{m_n}} \Sigma^{1/2} e_{i,m} + r_{i,n}, \quad \sup_{1 \le i \le n} \|r_{i,n}\| = o_p(m_n^{-1/2}).$$

Consequently,

$$\Delta_n = \frac{1}{n\sqrt{m_n}} \sum_{i=1}^{n} \nabla_Z s_i(\theta^*, Z_i^l)^\top \Sigma^{1/2} e_{i,m} + o_p(n^{-1/2}). \tag{C.3}$$

By the Lindeberg-Feller CLT and Condition C.5,

$$\sqrt{n} \, S_n(\theta^*) \xrightarrow{d} \mathcal{N}(0, \text{Var}[s_i(\theta^*, Z_i^{l,*})]).$$

Rewrite the leading term of (C.3) as

$$\sqrt{\frac{n}{m_n}} \left\{ \frac{1}{n} \sum_{i=1}^{n} \nabla_Z s_i(\theta^*, Z_i^l)^\top \Sigma^{1/2} e_{i,m} \right\}.$$

As $e_{i,m}$ are i.i.d. and independent of $(Y_i^h, X_i, Z_i^l)$, the inner average has mean $G\Sigma^{1/2} n^{-1} \sum e_{i,m} = 0$ and variance $n^{-1} G\Sigma G^\top$. Therefore, conditional on the data,

$$\sqrt{n} \, \Delta_n \xrightarrow{d} \sqrt{c} \mathcal{N}(0, \, G\Sigma G^\top),$$

and this limit is independent of $\sqrt{n} S_n(\theta^*)$ by construction. Adding the two independent Gaussian limits yields

$$\sqrt{n} \, \hat{S}_n(\theta^*) \xrightarrow{d} \mathcal{N}(0, U), \quad U := \text{Var}[s_i(\theta^*, w_i)] + c \, G\Sigma G^\top.$$

With the same arguments using the uniform LLN as in (C.1),

$$\hat{H}_n(\theta) = \frac{1}{n} \sum_{i=1}^{n} -\nabla_\theta^2 \ell_{n,i}(\theta; \hat{Z}_i^l, X_i) \xrightarrow{p} \mathcal{I}(\theta^*)$$

uniformly on a neighborhood of $\theta^*$. In particular, $\hat{H}_n(\bar{\theta}_n)^{-1} \xrightarrow{p} \mathcal{I}(\theta^*)^{-1}$ for any random $\bar{\theta}_n$ between $\hat{\theta}_n$ and $\theta^*$. Next, using a mean-value expansion of the firstorder condition $\hat{S}_n(\hat{\theta}_n) = 0$,

$$0 = \sqrt{n}\,\hat{S}_n(\theta^*) - \hat{H}_n(\bar{\theta}_n)\,\sqrt{n}(\hat{\theta}_n - \theta^*),$$

whence $\sqrt{n}(\hat{\theta}_n - \theta^*) = \hat{H}_n(\bar{\theta}_n)^{-1}\sqrt{n}\hat{S}_n(\theta^*)$. Combine the convergence of $\hat{H}_n(\bar{\theta}_n)^{-1}$ with the Gaussian limit of $\sqrt{n}\hat{S}_n(\theta^*)$,

$$\sqrt{n}(\hat{\theta}_n - \theta^*) \xrightarrow{d} \mathcal{I}(\theta^*)^{-1}\mathcal{N}(0, \Sigma) = \mathcal{N}\big(0,\ \mathcal{I}(\theta^*)^{-1}U\mathcal{I}(\theta^*)^{-1}\big).$$

Selecting the $(\beta, \gamma)$ block completes the proof of part (b).

$\square$

## D   Covariates

We construct two types of covariates from each LLM-generated response. The first type consists of LLM-scored covariates, derived using GPT-4o-mini based evaluations via prompts described in Appendix A.3. These are numeric ratings ranging from 0 to 5 and include aspects including coherence, factuality, clarity, conciseness, creativity, consistency, engagement, fluency, appropriateness, and sentiment. Each of these captures a subjective quality of the response, as detailed in Table 16. To obtain these scores, we extract the top 20 most probable responses along with their log-probabilities from the model output. Then, calculate a probability-weighted average of the probable scores to gain a stable response.

The second type comprises automatically computed lightweight covariates, which are extracted using rule-based or statistical methods without human prompting or query LLMs. These features cover five broad dimensions of a response: length (e.g., word and sentence counts), readability (e.g., Flesch Reading Ease score), style and formatting (e.g., paragraph structure, use of markdown), sentiment (e.g., TextBlob and VADER sentiment scores), and discourse markers counts (e.g., counts of additive or causal connectors). Table 17 provides variable names and descriptions for each of these lightweight covariates. Some of the constructed covariates exhibit high correlation with others. We address this using clustering methods, as discussed in the following section.

Table 16: LLM-scored covariates

| Variable | Description |
| --- | --- |
| coherence | how coherent the response is |
| factuality | how factually accurate the response is |
| clarity | how clear the language of the response is |
| conciseness | how concise the response is |
| creativiey | how creative and original the response is |
| consistency | how consistent the style and content the response is |
| engagement | how engaging the response is to the reader |
| fluency | how fluent the response reads |
| appropriateness | how appropriate the tone and style of the response is for the intended content |
| sentiment | how positive the overall emotional tone of the response is |

Table 17: Automatic lightweight covariates

| Variable | Description |
|---|---|
| *Length and Tokenization* | |
| response_word_count | Total number of word tokens in the response. |
| response_char_count | Total number of characters (including spaces and punctuation). |
| relative_char_count | Average characters per token (`response_char_count / response_word_count`). |
| response_sentence_count | Total number of detected sentences. |
| response_avg_sentence_length | Average tokens per sentence (`response_word_count / response_sentence_count`). |
| *Readability and Lexical Diversity* | |
| flesch_reading_ease | Flesch Reading Ease score (higher = easier to read) [9]. |
| flesch_kincaid_grade | Flesch-Kincaid gradelevel estimate [19]. |
| gunning_fog | Gunning Fog index (years of education needed) [13]. |
| smog | SMOG index emphasizing polysyllabic words [31]. |
| lexical_diversity | Number of unique word types [e.g., 16]. |
| relative_lexical_diversity | Type-token ratio [16, 30] (`lexical_diversity / response_word_count`). |
| *Style and Formatting* | |
| exclamation_count | Number of ! characters. |
| relative_exclamation_count | Exclamation marks per token. |
| question_count | Number of ? characters. |
| relative_question_count | Question marks per token. |
| paragraph_count | Number of paragraphs (nonempty newlineseparated blocks). |
| avg_paragraph_length | Average tokens per paragraph. |
| relative_paragraph_count | Paragraphs per token. |
| linebreak_count | Number of \n newline characters. |
| relative_linebreak_count | Newlines per token. |
| contains_code_block | 1 if a Markdown code block ("'...'") is present, else 0. |
| header_count | Count of Markdown headers (lines starting with #). |
| relative_header_count | Headers per token. |
| bold_count | Number of bold words (**bold** or __bold__). |
| relative_bold_count | Bold words per token. |
| italic_count | Number of italicized words (*italic* or _italic_, excluding bold). |
| relative_italic_count | Italic words per token. |
| list_count | Count of list items (-, *, or numbered markers). |
| relative_list_count | List items per token. |
| *Sentiment and Tonal Analysis* | |
| sentiment_polarity | TextBlob polarity score (-1 to 1) [7, 29]. |
| sentiment_subjectivity | TextBlob subjectivity score (0 to 1) [7, 29]. |
| sentiment_scores_comp | VADER compound sentiment score [10]. |
| sentiment_scores_pos | VADER positivesentiment proportion [10]. |
| *Discourse and Coherence* | |
| discourse_mk_add | Count of additive discourse markers (e.g., *furthermore*). |
| discourse_mk_con | Count of contrastive markers (e.g., *however*). |
| discourse_mk_cau | Count of causal markers (e.g., *therefore*). |
| discourse_mk_ex | Count of example markers (e.g., *for example*). |
| discourse_mk_sum | Count of summarizing markers (e.g., *overall*). |

### D.1 Clustering covariates

Algorithm 1 compresses a potentially large and collinear covariate set into smaller covariate groups that are less correlated. First, all columns of the design matrix $F$ are z-standardised. The dissimilarity matrix $D = 1 - \text{corr}(F^\top)$ is then subjected to a top-down complete linkage agglomerative procedure: Starting with clusters $p-1$ and progressively merging features, we repeatedly (i) partition the covariates, (ii) extract the first principal component from each cluster, thus summarizing its shared signal in a single latent factor, and (iii) check whether any pair of resulting factors still exhibits an absolute correlation above the user-specified threshold $\tau$. The loop terminates at the coarsest partition whose principal components are mutually "sufficiently uncorrelated," ensuring that the final matrix $\widehat{F}$ contains compact, approximately independent summaries of the original features. A final z-score standardisation puts these latent factors on a common scale, making them immediately suitable for downstream modelling or inference.

In Tables 18 and 19, we describe the resulting covariate clusters for BGB and CA based on the initial covariates described in Tables 16 and 17.

---

**Algorithm 1** Covariate clustering

---

**Require:** Covariate matrix $F \in \mathbb{R}^{n \times p}$; correlation threshold $\tau$ (default 0.7)
1: Standardise $F$ column-wise using z-scores.
2: $D \leftarrow 1 - \text{corr}(F^\top)$              ▷ feature-wise dissimilarity matrix
3: **for** $k \leftarrow p-1, p-2, \ldots, 1$ **do**            ▷ top-down search
4:      Apply complete-linkage agglomerative clustering on $D$ to obtain $k$ clusters with labels $\ell_1, \ldots, \ell_p$.
5:      Initialise $\widehat{F} \leftarrow [\,]$.
6:      **for all** cluster $c \in \{1, \ldots, k\}$ **do**          ▷ per-cluster PCA
7:          Fit PCA on the training columns with label $c$; keep first principal component $\mathbf{z}^{(c)}$.
8:          Append $\mathbf{z}^{(c)}$ to $\widehat{F}$.
9:      **end for**
10:      $\Sigma \leftarrow \text{corr}(\widehat{F}^\top)$; set $\Sigma_{ii} \leftarrow -99$ for all $i$.
11:      **if** $\max(\Sigma) < \tau$ **then**             ▷ stopping rule
12:          **break**
13:      **end if**
14: **end for**
15: Standardise $\widehat{F}$ column-wise using z-scores.
16: **return** $\widehat{F}$.

---

Table 18: Cluster definitions for BigGen Bench

| Cluster | Description |
|---------|-------------|
| Writing Quality | Composite rating combining coherence, clarity, consistency, fluency, and appropriatenessbroadly reflecting how well-structured, clear, and appropriate a response is ('coherence', 'clarity', 'consistency', 'fluency', 'appropriateness'). |
| Text Length | Basic length and diversity features, including word, character, and sentence counts, as well as the count of unique word types ('response_word_count', 'response_char_count', 'response_sentence_count', 'lexical_diversity'). |
| Italics | Frequency of italicized text in markdown, measured as both the number of italic words and as a proportion of tokens ('italic_count', 'relative_italic_count'). |
| Bold Text | Usage of bold markdown formatting, captured as both total count and per-token frequency ('bold_count', 'relative_bold_count'). |
| Lists | Frequency and density of list items, including both the absolute number and relative count per token ('list_count', 'relative_list_count'). |
| Headers | Markdown header usage, including total header count and density per token ('header_count', 'relative_header_count'). |
| Creativity/Engagement | Ratings for creativity and reader engagement, reflecting how original and captivating the response is ('creativity', 'engagement'). |
| Positive Sentiment | Proportion of content flagged as positive sentiment by VADER ('sentiment_scores_pos'). |
| Conciseness | Evaluation of how brief and to-the-point the response is ('conciseness'). |
| Contrast Markers | Frequency of contrastive discourse cues such as however and yet ('discourse_mk_con'). |
| Layout Density | Density of structural elements, capturing relative number of paragraphs and line breaks per token ('relative_paragraph_count', 'relative_linebreak_count'). |
| Causal Markers | Frequency of causal discourse signals like thus or therefore ('discourse_mk_cau'). |
| Structure Counts | Raw counts of paragraphs and explicit line breaks ('paragraph_count', 'linebreak_count'). |
| Sentiment | Overall sentiment rating, typically on a 05 scale ('sentiment'). |
| Readability Grade | Grade-level readability indices, such as Flesch-Kincaid, Gunning Fog, and SMOG, that estimate years of education needed ('flesch_kincaid_grade', 'gunning_fog', 'smog'). |
| Exclamation Density | Exclamation marks per token, reflecting the relative use of ! for emphasis ('relative_exclamation_count'). |
| Readability Ease | Flesch Reading Ease score, with higher values indicating easier reading ('flesch_reading_ease'). |
| Polarity | Sentiment polarity from TextBlob, ranging from 1 (negative) to +1 (positive) ('sentiment_polarity'). |
| Question Density | Relative frequency of question marks, indicating tendency to ask questions ('relative_question_count'). |
| Paragraph Length | Average number of tokens per paragraph, reflecting structural granularity ('avg_paragraph_length'). |
| Code Block | Binary indicator for the presence of markdown code blocks ("'..."') ('contains_code_block'). |
| Additive Markers | Frequency of additive discourse markers such as furthermore or moreover ('discourse_mk_add'). |
| Summary Markers | Frequency of summarizing discourse cues such as overall or in conclusion ('discourse_mk_sum'). |
| Character Density | Average number of characters per word token ('relative_char_count'). |
| Exclamation Count | Total number of exclamation marks in the response ('exclamation_count'). |
| Lexical Ratio | Relative lexical diversity, measured as the type-token ratio ('relative_lexical_diversity'). |
| Subjectivity | Degree of subjectivity (0 = objective, 1 = subjective) as measured by sentiment analysis ('sentiment_subjectivity'). |
| Sentence Length | Average tokens per sentence, reflecting sentence complexity ('response_avg_sentence_length'). |
| Example Markers | Frequency of example-giving discourse markers such as for example ('discourse_mk_ex'). |
| Compound Sentiment | VADER compound sentiment score summarizing overall tone ('sentiment_scores_comp'). |
| Question Count | Absolute number of question marks ('question_count'). |

Table 19: Cluster definitions for Chatbot Arena

| Cluster | Description |
| --- | --- |
| Text Length | Core size measures of the responseword, character, and sentence countsplus raw lexical diversity (`response_word_count`, `response_char_count`, `response_sentence_count`, `lexical_diversity`). |
| Creativity/Engagement | Ratings capturing originality and reader engagement (`creativity`, `engagement`). |
| Readability Grade | Grade-level readability indices, including Flesch-Kincaid, Gunning Fog, and SMOG (`flesch_kincaid_grade`, `gunning_fog`, `smog`). |
| Consistency | Measures how consistent the style and content of the response is (`consistency`). |
| Bold Text | Use of bold markdown formatting, both absolute and relative (`bold_count`, `relative_bold_count`). |
| Causal Markers | Frequency of causal discourse cues (e.g., thus, therefore) (`discourse_mk_cau`). |
| Language Quality | Ratings related to clarity, fluency, and appropriateness of the response (`clarity`, `fluency`, `appropriateness`). |
| Example Markers | Frequency of example cues (e.g., for example, for instance) (`discourse_mk_ex`). |
| Conciseness | Assessment of brevity and avoidance of unnecessary verbosity (`conciseness`). |
| Structure Counts | Absolute counts of paragraphs and explicit line breaks (`paragraph_count`, `linebreak_count`). |
| Additive Markers | Frequency of additive discourse cues (e.g., furthermore, moreover) (`discourse_mk_add`). |
| Polarity | Polarity score from sentiment analysis (TextBlob; -1 to 1) (`sentiment_polarity`). |
| Contrast Markers | Frequency of contrastive discourse cues (e.g., however, yet) (`discourse_mk_con`). |
| Sentiment | Overall sentiment score (`sentiment`). |
| Coherence | Measures how coherent the response is (`coherence`). |
| Linebreak Density | Relative number of newlines per token (`relative_linebreak_count`). |
| Lists | Absolute count of list items (`list_count`). |
| Subjectivity | Degree of subjectivity in sentiment analysis (0 = objective, 1 = subjective) (`sentiment_subjectivity`). |
| Readability Ease | Flesch Reading Ease score (higher values = easier to read) (`flesch_reading_ease`). |
| Paragraph Length | Average number of tokens per paragraph (`avg_paragraph_length`). |
| Code Block | Binary indicator for the presence of fenced code blocks (`contains_code_block`). |
| Question Density | Question marks per token (relative measure) (`relative_question_count`). |
| List Density | List items per token (`relative_list_count`). |
| Exclamation Count | Total number of exclamation marks (`exclamation_count`). |
| Paragraph Density | Relative number of paragraphs per token (`relative_paragraph_count`). |
| Character Density | Average characters per token (`relative_char_count`). |
| Count Italic | Absolute number of italicized words (`italic_count`). |
| Question Count | Absolute number of question marks (`question_count`). |
| Positive Sentiment | Proportion of text flagged as positive by VADER (`sentiment_scores_pos`). |
| Compound Sentiment | VADER compound sentiment score summarizing overall tone (`sentiment_scores_comp`). |
| Summary Markers | Frequency of summarizing discourse cues (`discourse_mk_sum`). |
| Lexical Ratio | Relative lexical diversity; type-token ratio (`relative_lexical_diversity`). |
| Relative Italic | Italicized words per token (`relative_italic_count`). |
| Exclamation Density | Exclamation marks per token (`relative_exclamation_count`). |
| Header Density | Headers per token (`relative_header_count`). |
| Headers | Absolute count of Markdown headers (`header_count`). |
| Sentence Length | Average tokens per sentence (`response_avg_sentence_length`). |

# E   Model extensions

## E.1   Modeling the expected judgement

This subsection is especially useful when $Y^h$ and $Y^l$ are ordinal with many possible values or continuous (*i.e.*, could possibly assume any value inside a closed interval). Let us take $Y^h, Y^l \in [0, K]$. We assume

$$\mathbb{E}[Y^h \mid I, O] = K\sigma(Z^h) \text{ and } \mathbb{E}[Y^l \mid I, O] = K\sigma\left(Z^l\right) \text{ where } Z^l \triangleq \alpha + \beta Z^h + \gamma^\top X.$$

In this setup, $\mathbb{E}[Y^l \mid I, O]$ can be computed exactly using the log probabilities or estimated using sampling. Take $Z^l = \sigma^{-1}(\mathbb{E}[Y^l \mid I, O]/K)$. Then,

$$\mathbb{E}[Y^h \mid I, O] = K\sigma\left(-(1/\beta)\alpha + (1/\beta)Z^l - (1/\beta)\gamma^\top X\right),$$

and, equivalently,

$$Y^h = K\sigma\left(-(1/\beta)\alpha + (1/\beta)Z^l - (1/\beta)\gamma^\top X\right) + \varepsilon.$$

For an error term $\varepsilon$ with $\mathbb{E}[\varepsilon \mid I, O] = 0$. The model can be fitted using non-linear least squares using human labels. The asymptotic distribution of the estimators could be derived using the theory of M-estimation; the bootstrap could also be used for approximating it.

## E.2   Categorical/Multinomial judgements

An extension of the ordinal model assumes $Y^h, Y^l \in \{0, \cdots, K\}$, but where no ordering is needed; we rely on the multinomial logistic regression model [45]. This model can still be used in the case of ordered outputs, but it is harder to interpret. For this formulation, we assume

$$\mathbb{P}(Y^h = k \mid I, O) = \begin{cases} \frac{1}{1+\sum_{j=1}^K \exp(Z^{h(j)})} & \text{if } k = 0, \\ \frac{\exp(Z^{h(k)})}{1+\sum_{j=1}^K \exp(Z^{h(j)})} & \text{if } 1 \leq k \leq K, \end{cases}$$

where the class $0$ is the base class; in practice, the practitioner could use any other class as the base one. As before, we assume a connection between human and LLM judgements; in this case, we have

$$Z^{l(j)} \triangleq \alpha_j + \beta_j Z^{h(j)} + \gamma_j^\top X.$$

In this formulation, we can obtain $Z^{l(j)}$ by using the formula

$$Z^{l(j)} = \log \frac{\mathbb{P}(Y^l = k \mid I, O)}{\mathbb{P}(Y^l = 0 \mid I, O)},$$

and then fit the model, using human labels, by expressing $Z^{h(j)}$ in terms of $Z^{l(j)}$ and maximizing the loglikelihood with respect to the model parameters. The asymptotic distribution of the estimators could be derived using the theory of maximum likelihood estimation; the bootstrap could also be used for approximating it.

