# OpenReview forum: "Bridging Human and LLM Judgments: Understanding and Narrowing the Gap"
_NeurIPS.cc/2025/Conference — NeurIPS 2025 poster_

### Official Review · Reviewer_DPsg · 2025-07-01

**Clarity:** 2
**Significance:** 3
**Originality:** 3
**Rating:** 5
**Confidence:** 2

**Summary:**

This paper proposes a statistical model to analyze the discrepancies between LLM judges' outputs (in the form of scores or comparisons) and human evaluations. Based on this model, the authors design a calibration method for LLM judges and conduct hypothesis testing on various covariates used in LLM evaluations. A notable and surprising conclusion drawn from the analysis is that *LLM judges favor brevity relative to human raters*, which stands in contrast to prior findings.

**Questions:**

1. Line 133: The authors state that using reasoning steps biases the computed probabilities. Could the authors clarify why this is the case? It seems the main challenge is technical—namely, extracting token probabilities step-by-step—which, while complex, may not necessarily introduce bias.
2. Line 160: Should the model-implied probabilities be conditioned on $\hat{Z}^l$ rather than $\hat{Z}^h$?
3. Line 180: Consider providing a more intuitive explanation of what rejecting $H_0: \gamma_j = 0$ means in the context of the model.
4. Line 244: Why are log probabilities used for GPT-4o-mini while CoT is used for the other judges? A clearer justification for this choice would help readers understand the experimental consistency.
5. Line 257: Does “probabilistic classifier” here refer to the fitted model? If so, it would be helpful to make this explicit.
6. The multinomial version of the model is placed in the appendix, but given its brevity, it might be better included in the main text to maintain continuity.
7. The implementation details of the second baseline ("LogReg") are unclear. How exactly is it constructed, and how does it differ from the proposed model? A more thorough comparison is needed.
8. The caption for Figure 1 is incorrectly labeled as Table 1.
9. According to Figure 1, many results have large standard deviations—much larger than the mean differences between LogReg and the proposed model—raising questions about the robustness of the results.
10. Line 303: This claim that conflicts with Dubois et al. is that Dubois et al. found LLM judges favor verbosity, not that length-controlled scoring improves human alignment, so the sentence should be revised for accuracy. Furthermore, although Table 2 shows consistent trends across four models, none of them match those used in Dubois et al. To strengthen this point, it would be valuable to replicate the analysis on overlapping models.

**Ethical Concerns:**

["NO or VERY MINOR ethics concerns only"]

**Final Justification:**

The author's reply helped me better understand some technical details: LogReg is a naive version of the model proposed in the paper. Through my communication with the author, I gained a clearer understanding of the rationale behind their conclusion on length bias, which significantly differs from previous research.

**Limitations:**

yes

**Quality:**

3

**Strengths And Weaknesses:**

**Strengths**

* The paper is mathematically rich, technically sound, and methodologically coherent. The proposed statistical model supports a fitting algorithm, a calibration method, and covariate-based hypothesis testing, all grounded in solid mathematical foundations.
* It addresses a crucial issue—the biases of LLM judges—which directly impacts the reliability and fairness of LLM-based evaluations. Importantly, rather than attempting to eliminate all bias, the authors suggest that alignment with human biases may be acceptable, marking a thoughtful and novel departure from work that seeks to “over-correct” for biases.
* The paper reports several intriguing findings, most notably that LLM judges tend to prefer shorter responses relative to human raters. This challenges previous assumptions and adds valuable insight to the ongoing discourse.

**Weaknesses**

* The definition and scope of “bias” could benefit from further clarification. As acknowledged by the authors (Line 171), bias ideally refers to models relying on features unrelated to content quality. However, many of the covariates included in this study—especially those derived from LLM scores such as factuality—are closely tied to quality, which can lead to confusion. Although this is briefly discussed in the limitations section, a more in-depth explanation would strengthen the paper.
* Some experimental designs lack detail or appear underdeveloped. In particular, the use and interpretation of the “LogReg” baseline, and how it supports the claim that LLM judges favor brevity, need clearer justification and elaboration.

---

> ### Author Rebuttal · Authors · 2025-07-31
>
> Thank you for taking the time to review our paper. We have responded to your comments and questions below. If you have any further feedback or concerns, please feel free to let us know.
>
> -----------
>
> ### **The definition and scope of bias**
>
> We agree that our current phrasing around bias (Line 171) can be sharpened. In our framework, bias is any systematic deviation of an LLM judge’s rating from a benchmark rater and, in our experiments, we use human ratings as the gold standard. For instance, by including factuality scores as a covariate, we test whether an LLM judge’s deviations from true quality vary systematically with factuality. If so, it indicates a bias associated factuality, i.e.,  how the LLM judge prioritizes factual correctness relative to the benchmark rater. In the introduction of the paper, we will clarify our definition of bias as the conditional dependence of LLM ratings on any feature or attribute of an input/output given human preferences. Thank you for your suggestion.
>
> -----------
>
> ### **The use and interpretation of the "LogReg" baseline**
>
> The LogReg baseline can be seen as a naive version of our multinomial logistic regression method. In this baseline, we use the raw probabilities output by the LLM for each possible rating and fit a standard logistic regression to predict human ratings. While this method can perform reasonably well for predictive purposes, it lacks a principled modeling foundation and is not interpretable. In contrast, our multinomial approach directly models the relationship between the LLM's logits and human ratings, offering a clear interpretation as detailed in the appendix. It is also important to clarify that the LogReg baseline is only included in "Application 1" for completeness and comparison. It is not used to support the claim that LLM judges favor brevity; that claim is addressed using other analyses in the paper. Thanks for your comment; we will make this clearer in the paper.
>
> -----------
>
> ### **Questions**
>
> - **(Q1, line 133) Why using reasoning steps biases the computed probabilities:** This is the case because once the LLM has generated the reasoning steps (sequence of tokens $R$) it conditions its score on $R$, then biasing its ratings towards something that are more compatible with $R$. Ideally, we want to marginalize our extracted probabilities over all possible reasoning paths. That's why we sample many times to estimate these probabilities. In practice, we are using Monte Carlo integration to estimate the following sum
> $$
> \mathbb{P}(Y^l=k \mid I,O) = \sum_r\mathbb{P}(Y^l=k \mid I,O, R=r)  \mathbb{P}( R=r \mid I,O)
> $$
> and avoiding the use of individual $\mathbb{P}(Y^l=k \mid I,O, R=r)$'s. Please let us know if this does not make sense!
>
> - **(Q2, line 160) The model-implied probabilities $p_k(\hat\alpha_1,\ldots, \hat\alpha_K, \hat Z^h):$** This probability depends on $\hat Z^h$ following its definition on line 104. Here, we use $\hat Z^h$ to represent an estimated human preference latent factor calculated as $(1/\hat\beta)Z^l - (1/\hat\beta)\hat\gamma^\top X $, which is substituted in the definition of the  model-implied probability. We will clarify this in the revision.
>
> - **(Q3, line 180) A more intuitive explanation of what rejecting $H_0: \gamma_j=0$ means:** Thank you for this suggestion. Rejecting the null here means that, holding all other covariates constant, variation in feature $j$ of an output causes a statistically significant change in the gap between human and LLM ratings. Intuitively, this tells us that feature $j$ systematically shifts the LLM’s judgments relative to humans, in other words, it is a source of LLM judgment bias. We will add this clarification.
>
> - **(Q4, line 244) Why are log probabilities used for GPT-4o-mini while CoT is used for the other judges:** We chose to use CoT sampling for open-weight models because CoT improves the quality of those judges, and running these models locally makes CoT sampling computationally inexpensive. In contrast, generating CoT traces for GPT-4o-mini is prohibitively expensive since it relies on the paid API. For this reason, we use log probabilities for GPT-4o-mini instead. Ideally, we would have liked to sample CoT traces for GPT-4o-mini as well, but this was not feasible due to cost constraints. We will clarify this point in the revised text.
>
> - **(Q5, line 257) Does "probabilistic classifier" here refer to the fitted model:** Yes, by "probabilistic classifier" we mean any of the three methods "LogReg", "Multinomial", and "Ordinal".
>
> - **(Q6) Including the multinomial version of the model:** We will include the multinomial version in the main text.
>
> - **(Q7) The implementation details of the second baseline ("LogReg") are unclear:** We have responded to this point above.
>
> - **(Q8) The caption for Figure 1:** Thank you for pointing this out. We will revise accordingly.
>
> - **(Q9, Figure 1) Many results have large standard deviations and robustness of the results:** As mentioned above, the LogReg method is a naive version of our method and should not be viewed as a competitor (since it is not published anywhere). The real competitor is the "Raw" method and we clearly beat that one; please check our response to Reviewer uPE2 for a more detailed explanation and an extra experiment.
>
>
> - **(Q10, line 303) Clarify the the verbosity bias claim and replicate on shared models with Dubois et al:** Dubois al. shows in their paper and GitHub repository (AlpacaEval) that the length-controlled version of AlpaceEval correlates better with Chatbot Arena ELO scores and, because of that, they claim better alignment with human scores. In their repo, they explicitly claim that their new metric is "highly correlated with humans". To strenghten our point, we have run the same bias analysis for GPT-4-Turbo (main judge in AlpacEval) and show the negative bias towards length still holds:
>
> |             GPT-4-Turbo      | Text Length      |
> |:------------------------:|:---------------:|
> | BigGenBench              | **-0.49*****      |
> | Chatbot Arena            | **-0.90*****      |

---

> > ### Comment · Reviewer_DPsg · 2025-08-06
> >
> > Thank you for your detailed response! I'm still curious about how to explain the large discrepancy between your conclusions and those of Dubois et al. Do you think this is due to differences in aggregation methods? Specifically, Dubois et al. used system-level evaluation—aggregating the results of LLM-evaluation to derive model rankings and comparing them with the Chatbot Arena—whereas your experiment used instance-level evaluation results.

---

> > > ### Author Response · Authors · 2025-08-06
> > >
> > > We thank the reviewer for the engagement in discussion!
> > >
> > > That’s a great question. While we don’t have a definitive answer, we see a couple of likely reasons for the discrepancy:
> > >
> > > 1. **Evaluation Methodology:** As you mentioned, our approach directly compares LLM and human judgments at the instance level, while Dubois et al. aggregate LLM evaluation results at the system level to derive model rankings (e.g., comparing with Chatbot Arena ELO scores). This difference in aggregation introduces additional factors and potential confounders, making the results not directly comparable. *We think that working on the instance level is a more direct and reliable way of drawing such conclusions.*
> > >
> > > 2. **Magnitude of Correlation Differences:** As reported in their GitHub, the length-controlled metric increases the correlation with human judgments only marginally, from 0.94 to 0.98. Both values are already very high, so the significance of this improvement is unclear.

---

> > > > ### Comment · Reviewer_DPsg · 2025-08-07
> > > >
> > > > Thank you for your rapid reply! After careful consideration, I have raised my overall rating. I look forward to seeing our discussion reflected in subsequent versions.

---

### Official Review · Reviewer_uPE2 · 2025-07-01

**Clarity:** 3
**Significance:** 3
**Originality:** 3
**Rating:** 4
**Confidence:** 4

**Summary:**

- This paper introduces a statistical framework to model human and LLM judgements, which enables formal tests of human-LLM discrepancies.
- The authors evaluate the framework on two benchmarks (BigGen Bench and Chatbot Arena) and four LLM judges.
- The authors study two applications on improving LLM judgements with few annotations and identifying biases in LLM judges.

**Questions:**

- Can the authors evaluate more models as LLMs on even a subset of the questions? I’m interested in “profiling” different LLMs and seeing trends.
- Can the authors discuss the use of in-the-wild data and its potential implications?
- Can the authors clarify comments around both applications and their respective experimental designs?

**Ethical Concerns:**

["NO or VERY MINOR ethics concerns only"]

**Final Justification:**

The reviewers provided a nice set of follow-up experiments in their rebuttal. I look forward to them integrating these results into the new version of the paper.

**Limitations:**

Yes

**Quality:**

2

**Strengths And Weaknesses:**

Quality:
- There are only 4 LLM judges evaluated in this work. It would be great to see results on the two applications on a larger set of models, especially bigger / SOTA models.
- While it is great that the experiments are on in-the-wild data (e.g., Chatbot Arena), it is hard to know what the “ground truth” results should be. For example, with Application 2, we don’t know what the ground truth LLM biases are. It would be helpful to have some synthetic but more controlled experiments to show the effectiveness of the framework.

Clarity:
- The methodology is nicely written; my main comments are on the two applications.
- For Application 1, given the error bars, it’s not clear that the proposed methods improve accuracy. In fact, even with increasing sample size, the accuracy looks about the same. Could the authors provide a much larger sample size variant (e.g., an order or two magnitude greater) to compare against? Additionally, can the authors discuss how there might be significant variance in human annotators when given only a small number of annotations? How is that accounted for?
- For Application 2, how do results vary based on the amount of training data? How robust are the findings of observed LLM biases?
- Additionally, for Chatbot Arena experiments, did you filter for specific categories? The types of biases we might expect could vary depending on the topic.

Significance:
- This work contributes to an important and growing area of using LLMs as judges.

Originality:
- A formal approach to comparing human and LLM judges is very novel.

---

> ### Author Rebuttal · Authors · 2025-07-31
>
> Thank you for your work on our paper. We have addressed your comments and questions below. Please let us know if you have further questions or concerns.
>
> -----------
>
> ### **Results on bigger / SOTA models**
>
> We have run extra results on GPT-4.1 and GPT-4.1-nano, which are bigger or SOTA models (and affordable). Results are qualitatively similar to our previous models.
>
> **Results for Application 1:**
>
> Please see "Whether the proposed method improves accuracy" point below, where we report full results for accuracy (in all other metrics, our method still beats Raw by a large margin).
>
> **Results for Application 2:**
>
> | BigGenBench              | GPT 4.1 nano | GPT 4.1    |
> |----------------------|--------------|------------|
> | Writing Quality      | **-0.38*****   | -0.10      |
> | Text Length          | **-0.83*****   | **-0.39***** |
> | Positive Sentiment   | **-0.31*****   | **-0.12*** |
> | Layout Density       | -0.23        | **-0.15*** |
> | Causal Markers       | **-0.19****    | -0.09      |
> | Structure Counts     | **0.35*****    | **0.16***  |
> | Sentiment            | **0.24****     | 0.10       |
> | Code Block           | **0.20****     | 0.07       |
> | Compound Sentiment   | **0.27*****    | 0.06       |
>
>
> | Chatbot Arena            | GPT 4.1 nano   | GPT 4.1    |
> |--------------------|---------------|------------|
> | Text Length        | **-2.05*****    | **-0.54***** |
> | Creativity         | **-1.27*****    | **-0.32***** |
> | Bold Text          | **0.74****      | **0.25*****  |
>
> -----------
>
> ### **Controlled experiments to show the effectiveness of the framework**
>
> We have conducted two additional controlled experiments to further validate our method, both of which use semi-synthetic setups that are more realistic than purely artificial data.
>
> 1. **First experiment:** We use GPT-4o-mini to simulate human ratings on BigGenBench queries. Next, we run GPT-4o-mini again, this time artificially biasing its latent scores $Z^l$ to *disfavor* specific markdown features—namely, bold/italicized words, headers, and lists. For each markdown feature (corresponding to the covariates $X_j$, $j=1,2,3$), we bias $Z^l$ by subtracting $X_j$ (one at a time). We then estimate $\gamma=(\gamma_1,\gamma_2,\gamma_3)$ with our method. In this controlled setting, biasing toward feature $j$ should result in $\gamma_j=-1$ while $\gamma_i=0$ for $i\neq j$. Standard errors are shown in parentheses.
>
> | Setting     | $\gamma_1$ (SE) | $\gamma_2$ (SE) | $\gamma_3$ (SE) |
> | ----------- | ------------------ | ------------------ | ------------------ |
> | no bias     | -0.26 (0.21)       | -0.36 (0.35)       | -0.17 (0.16)       |
> | bold/italic | -1.26 (0.21)       | -0.36 (0.35)       | -0.17 (0.16)       |
> | headers     | -0.26 (0.21)       | -1.36 (0.35)       | -0.17 (0.16)       |
> | lists       | -0.26 (0.21)       | -0.36 (0.35)       | -1.17 (0.16)       |
>
> 2. **Second experiment:** We conduct a slightly less controlled experiment by prompting GPT-4o-mini to give lower scores to responses containing each markdown feature, one at a time. Since this manipulation is done through prompting rather than direct latent score adjustment, the resulting biases are not as clean; for example, the LLM tends to be consistently biased against lists. Still, the results largely follow the expected direction.
>
> | Setting     | $\gamma_1$ (SE) | $\gamma_2$ (SE) | $\gamma_3$ (SE) |
> | ----------- | ------------------ | ------------------ | ------------------ |
> | no bias     | -0.255 (0.208)     | -0.362 (0.345)     | -0.167 (0.161)     |
> | bold/italic | -1.992 (0.290)     | 0.466 (0.486)      | -1.583 (0.234)     |
> | headers     | -0.069 (0.427)     | -1.014 (0.713)     | -3.061 (0.352)     |
> | lists       | -0.172 (0.319)     | -0.229 (0.536)     | -5.660 (0.260)     |
>
> These experiments demonstrate that our framework can recover the direction and magnitude of induced biases, both in tightly controlled and more realistic, prompt-based scenarios.
>
>
> -----------
>
> ### **Whether the proposed method improves accuracy**
>
> We repeated the analysis including $n=1280$ as well as both GPT-4.1-nano and GPT-4.1. The table below reports the **average improvement in accuracy** (of the “Ordinal” method over “Raw”) across random seeds, with standard deviations in parentheses. For almost all LLMs, we observe a substantial improvement in accuracy. In a few cases, the gains are within one standard deviation, so we cannot draw a firm conclusion. Taken as a whole, the results show a consistent positive trend in accuracy improvements across LLMs. The gains are still more impressive if we consider cross-entropy loss and calibration metrics though.
>
> **BigGenBench**
>
> | LLM/Sample size             | 20             | 40             | 80             | 160            | 1280           |
> | --------------- | -------------- | -------------- | -------------- | -------------- | -------------- |
> | GPT-4.1-nano    | 0.021 (0.043)  | 0.034 (0.026)  | 0.041 (0.021)  | 0.047 (0.021)  | 0.059 (0.016)  |
> | GPT-4.1         | -0.030 (0.021) | -0.024 (0.039) | -0.021 (0.022) | -0.017 (0.016) | -0.013 (0.017) |
> | GPT-4o-mini     | -0.024 (0.038) | -0.031 (0.044) | 0.000 (0.013)  | 0.004 (0.016)  | 0.010 (0.011)  |
> | LLaMa-3.1-8B-It | -0.001 (0.026) | 0.012 (0.020)  | 0.015 (0.021)  | 0.029 (0.019)  | 0.029 (0.020)  |
> | Selene-1-Mini   | 0.027 (0.035)  | 0.049 (0.022)  | 0.055 (0.023)  | 0.057 (0.019)  | 0.066 (0.021)  |
> | Prometheus-v2   | -0.008 (0.021) | 0.007 (0.022)  | 0.017 (0.018)  | 0.018 (0.019)  | 0.033 (0.017)  |
>
> **Chatbot Arena**
>
> | LLM/Sample size             | 20             | 40             | 80             | 160            | 1280          |
> | --------------- | -------------- | -------------- | -------------- | -------------- | ------------- |
> | GPT-4.1-nano    | -0.006 (0.022) | -0.005 (0.016) | -0.004 (0.015) | -0.005 (0.017) | 0.009 (0.013) |
> | GPT-4.1         | -0.024 (0.025) | -0.033 (0.037) | -0.015 (0.020) | -0.017 (0.019) | 0.001 (0.014) |
> | GPT-4o-mini     | 0.000 (0.017)  | -0.004 (0.019) | 0.003 (0.021)  | 0.013 (0.023)  | 0.030 (0.012) |
> | LLaMa-3.1-8B-It | 0.009 (0.022)  | 0.007 (0.027)  | 0.017 (0.026)  | 0.024 (0.019)  | 0.034 (0.017) |
> | Selene-1-Mini   | -0.013 (0.027) | -0.005 (0.021) | 0.014 (0.019)  | 0.007 (0.027)  | 0.034 (0.015) |
> | Prometheus-v2   | 0.016 (0.034)  | 0.002 (0.035)  | 0.006 (0.028)  | 0.032 (0.016)  | 0.033 (0.014) |
>
> -----------
>
> ### **Significant variance among human ratings**
>
> Some of this variance is captured by modeling the probability distribution of $Y^h$, since the same input-output pair $(I, O)$ can lead to a range of human ratings. An alternative approach would be to explicitly model $Z^h$ as a random variable, even after conditioning on $(I, O)$. However, this would considerably increase the model’s complexity, so we chose not to pursue it for now. Exploring this idea is a promising direction for future work though.
>
> -----------
>
> ### **For Application 2, how do results vary based on the amount of training data?**
>
> To assess the robustness of our method in detecting LLM biases, we repeat our analysis using varying random fractions of the available data. We set a significance threshold of 10\% (i.e., p-values below 0.10 are considered significant) and, for each LLM, treat the detection of nonzero $\gamma_j$ coefficients as a binary classification problem (reject vs. not reject the null hypothesis). The "ground truth" label is determined using the full dataset. For each data fraction, we evaluate how well the method predicts these significance decisions by reporting precision, recall, and accuracy. Our results (see tables below) show that, for both BigGenBench and Chatbot Arena, strong precision and accuracy can be achieved with as little as 50\% of the data. Recall is more challenging to improve, indicating that some biases may require more data to detect reliably.
>
> **Significance Prediction (BigGenBench)**
>
> | \% Data | Precision | Recall | Accuracy |
> | ------ | --------- | ------ | -------- |
> | 10     | 0.67      | 0.14   | 0.56     |
> | 25     | 0.63      | 0.23   | 0.57     |
> | 50     | 0.90      | 0.60   | 0.78     |
> | 75     | 0.90      | 0.79   | 0.86     |
> | 100    | 1.00      | 1.00   | 1.00     |
>
> **Significance Prediction (Chatbot Arena)**
>
> | \% Data | Precision | Recall | Accuracy |
> | ------ | --------- | ------ | -------- |
> | 10     | 0.33      | 0.08   | 0.75     |
> | 25     | 0.92      | 0.38   | 0.85     |
> | 50     | 0.61      | 0.46   | 0.80     |
> | 75     | 0.93      | 0.69   | 0.92     |
> | 100    | 1.00      | 1.00   | 1.00     |
>
> Overall, these results suggest that our method for detecting LLM biases is quite robust, with high precision and accuracy even when only half of the data is used. Recall improves with larger data fractions, highlighting the benefit of more data for sensitivity to weaker effects.
>
> -----------
>
> ### **For Chatbot Arena experiments, did you filter for specific categories**
>
> Please see our last point to Reviewer gh7b.
>
> -----------
>
> ### **The use of in-the-wild data (e.g., Chatbot Arena) and its potential implications**
>
> Thank you for this point. Our experiments leverage observational Chatbot Arena data to capture real-world usage, which brings both strengths and limitations:
>
> 1. In-the-wild data reflect the diversity and unpredictability of real user-LLM interactions, ensuring our findings generalize beyond narrow lab settings.
>
> 2. Because the data are not from a controlled trial, our estimated $\gamma_j$ parameters should be viewed as descriptive measures of how feature $j$ relates to human-LLM rating gaps, not as causal effects. Confounding variables and selection biases may influence these associations.

---

> > ### Comment · Reviewer_uPE2 · 2025-08-05
> >
> > Thank you to the authors for their additional results. I think this helps strengthen the work and I look forward to seeing it incorporated into the final version. I have updated my rating accordingly.

---

### Official Review · Reviewer_pkTY · 2025-07-03

**Clarity:** 3
**Significance:** 2
**Originality:** 3
**Rating:** 4
**Confidence:** 3

**Summary:**

The authors propose an ordered-logit model that ties human ratings to LLM-judge ratings through a shared latent "human-preference" score and a linear covariate term capturing systematic judge biases. They introduce a "logit trick" that recovers the latent scores without ever observing them, prove asymptotic normality for all key parameters, and demonstrate three practical uses: (i) probabilistic calibration of LLM judges, (ii) few-label alignment with humans, and (iii) hypothesis-driven detection of judge biases. Experiments on BigGen Bench and Chatbot Arena show lower cross-entropy, tighter calibration, better accuracy, and interpretable bias coefficients (e.g., length, creativity) than raw or logistic-regression baselines.

**Questions:**

See my feedback above.

**Ethical Concerns:**

["NO or VERY MINOR ethics concerns only"]

**Final Justification:**

Thank you for your response and clarification. While I believe the work is solid, I still have some reservations about the broader applicability of the proposed methods in today's landscape. I will be maintaining my current score.

**Limitations:**

Limitations are addressed in the discussion and corresponding questions are asked in my review.

**Paper Formatting Concerns:**

Minor typos: ("equilavently", "It represent").

**Quality:**

2

**Strengths And Weaknesses:**

I find the paper’s overall quality and clarity strong, and I especially appreciate the rigorous derivations presented in Proposition 3.1 and Theorem 3.2. Nevertheless, I have several concerns.

First, I am unsure how sensitive the method is to the choice of covariates. If handcrafted features were replaced by GPT-based embeddings, would the estimated γ coefficients shrink, explode, or remain stable? Quantifying this effect would illuminate how interpretable and trustworthy the bias term truly is.

Second, the discussion acknowledges that the model may break down when the latent-score assumption is violated, yet I see no empirical test of this vulnerability. Have you simulated scenarios in which the true relationship between human and LLM scores is nonlinear? A targeted robustness study would increase my confidence in the method.

My broader worry centers on the significance of this work in today’s LLM-evaluation landscape. As language models tackle ever-more complex tasks, obtaining high-quality "gold" human annotations is increasingly difficult, and recent studies highlight substantial disagreement among annotators on nuanced questions. In light of this, can human labels still be considered a reliable gold standard, and is narrowing the gap between humans and LLM judges still the most pressing problem?

Relatedly, your approach assumes discrete rating labels. How might it extend to settings where evaluation feedback is open-ended and expressed in free-form natural language rather than fixed scales?

---

> ### Author Rebuttal · Authors · 2025-07-31
>
> Thank you for your thoughtful review of our paper. We have addressed your comments and questions below. Please let us know if you have any additional questions or concerns.
>
> ------------------
>
> ### **Quantifying how sensitive the method is to the choice of covariates (point on GPT embeddings)**
>
> Our approach relies on logistic regression, a model class that is known to be robust to high-dimensional covariates, provided sufficient data and appropriate regularization are used. For example, if we were to use high-dimensional uninterpretable representations such as GPT embeddings as features, as you suggest, the standard practice would be to apply $l_2$ regularized logistic regression, as this would shrink the coefficients to a point where the method does not overfit the data. However, we have not taken this direction (of using embeddings), as it would substantially reduce the interpretability of our results, a key priority for us. We appreciate this suggestion and will add a discussion on these trade-offs in the paper. Thank you.
>
> ------------------
> ### **Vulnerability to misspecification**
>
> When our model is used for prediction, misspecification is not a significant concern; much of machine learning relies on models that are not exactly correct. If our focus is on statistical inference, the model can still be valuable for uncovering biases in LLM judgments, provided the misspecification is not too severe. The linear predictor we use is quite flexible, as we can include any basis functions of $X$ as covariates (and then capture nonlinear relationships).
>
> Empirically, we present below a simple simulation in which we introduce a mild nonlinearity into the LLM’s latent score generation to test robustness to misspecification, both for prediction and inference. We draw $Z^h_i$ from a Normal distribution $N(0,1)$ and sample $Y^h_i$ from $\mathrm{Categorical}\bigl(p(\alpha, Z^h_i)\bigr)$. We then set
> $$
> Z^l_i = \beta Z^h_i + \gamma^\top X_i + \delta\bigl(\gamma^\top X_i\bigr)^2,
> $$
> where $X_i$ is drawn from a multivariate Normal $N(0,I_3)$ distribution and $\delta$ takes values in $\\{0, 0.1, 0.25, 0.5,1,5\\}$, controlling the degree of quadratic distortion. We set $\beta=1$, $\gamma=(1,1,1)$, and $\alpha=(-1,1)$ LLM judgments $Y^l_i$ are sampled from the usual ordered-logit link $p(\eta, Z^l_i)$, and we fit our original linear model, assuming $Z^l_i = \beta Z^h_i + \gamma^\top X_i$. By comparing the estimated parameters $(\hat\beta, \hat\gamma,\hat{Z}^h, \mathbb{P}(Y^h=k|I,O))$ to their true values in terms of MAE as $\delta$ increases, we directly measure the impact of model misspecification. The results below show that we can still recover $\gamma$ (then at least we know approximately how big are the main effects; main channel of biases) and predict $\mathbb{P}(Y^h=k|I,O)$ with high accuracy, even under moderate misspecification. We will add this experiment to the paper. Interetingly, the most afected results are the ones for $\beta$ and $Z^h$ which is of less interest.
>
> **MAE**
>
> | $\delta$ | $\beta$ | $\gamma$ | $Z^h$ | $\mathbb{P}(Y^h=k\mid I,O)$  |
> |:--------:|:-------:|:--------:|:-----:|:---------------------------:|
> |   0      | 0.010   | 0.014    | 0.014 | 0.002                      |
> |  0.1     | 0.024   | 0.014    | 0.072 | 0.010                      |
> |  0.25    | 0.047   | 0.016    | 0.169 | 0.025                      |
> |  0.5     | 0.266   | 0.017    | 0.340 | 0.045                      |
> |   1      | 0.960   | 0.011    | 0.563 | 0.077                      |
> |   5      | 24.958  | 0.345    | 0.789 | 0.117                      |
>
>
>
> ------------------
>
> ### **The significance of this work in today’s LLM-evaluation landscape**
>
> Thank you for highlighting this broader concern. We fully recognize that as LLM tasks become more sophisticated, achieving a truly reliable "gold standard" through human annotation is increasingly difficult. However, our framework is flexible by design and is not tied to a single definition of the gold standard. Whether the benchmark consists of individual human judgments, a consensus among multiple annotators, or even alternative proxies, our approach is meant to detect and reduce inconsistencies between whatever reference is chosen and LLM assessments. In other words, as the notion of the "gold standard" evolves, our method remains applicable. Moreover, aligning LLMs with human preferences and judgments will continue to be important for many real-world applications, especially where trust, safety, and social acceptance matter. Since high-quality human annotation is expensive and slow, developing a clear understanding of the differences and similarities between human and LLM judgments remains a key challenge, regardless of how the benchmarks themselves shift over time.
>
> ------------------
> ### **Extension to open-ended and natural language evaluations**
>
> We appreciate the insight that various forms of evaluation feedback could be important. A proper way of doing this requires another paper by itself and we defer this point to future work; however we outline some possibilities and add a discussion to our paper. The core ideas should still be aligning human and LLM judgements in a shared representation and detecting systematic divergences.
>
> - A relatively direct way is to use prompt-based or lightweight classifiers to extract attributes (such as sentiment, critique categories, and overall rating) from free-form text. We then treat those attributes as pseudo-labels for calibration and bias detection the same way as in our current work.
>
> - Another route could be to use high-dimensional semantic representations (i.e., using embedding models) for human and LLM judgements, and then align those embedding vectors rather than scaler labels, detecting biases.

---

> > ### Comment · Area_Chair_RAt8 · 2025-08-05
> >
> > Dear Reviewer pkTY,
> >
> > The authors have provided a response to your review. Could you please let us know if their response has addressed your concerns? This step is mandatory.
> >
> > Thank you,
> > Area Chair

---

> > ### Comment · Reviewer_pkTY · 2025-08-05
> > **Thank you.**
> >
> > Thank you for your response and clarification. While I believe the work is solid, I still have some reservations about the broader applicability of the proposed methods in today's landscape. I will be maintaining my current score.

---

### Official Review · Reviewer_gh7b · 2025-07-03

**Clarity:** 4
**Significance:** 3
**Originality:** 3
**Rating:** 5
**Confidence:** 3

**Summary:**

The authors propose a statistical framework to align LLM agent ratings with human preferences. They model both human and LLM judgements as outcomes of a shared latent human preference signal. The LLM-as-a-judge framework is widely used and this work provides a solution to improve alignment with human judges.

**Questions:**

- Will your framework be publicly available after the review process?
- Did your method reveal any patterns in which LLMs (e.g., size or training regimen) are more misaligned with humans?

**Ethical Concerns:**

["NO or VERY MINOR ethics concerns only"]

**Final Justification:**

As I was already convinced that this paper is solid and should be accepted for the conference, I will not change my score.

**Limitations:**

yes

**Quality:**

3

**Strengths And Weaknesses:**

Strengths:
- The paper addresses a critical gap in recent research
- The paper is well written and structured
- The authors demonstrate a practical benefit of their approach, even with only a few labeled data points

Weaknesses:
- As the authors themselves address, the model is vulnerable to misspecification
- Necessary domain knowledge makes the framework less generalizable
- Human judgement are treated as gold standard, even if they can also contain biases. However it is the goal of the paper and also in most research studies to align agent with human judgements which is clearly achieved here

Despite the presence of weaknesses, this paper makes an important contribution to the formal alignment of human and LLM judgements, with implications for various applications of the LLM-as-judge framework.

Comments:
- You can use ~ before \citet or \citep to avoid references at the beginning of a line (see line 103)
- "Table 1" looks more like a Figure to me?

---

> ### Author Rebuttal · Authors · 2025-07-31
>
> Thank you for your work on our paper! We have addressed your comments and questions below. Please let us know if you have further questions or concerns.
>
> -------
> ### **Comments**
>
> - **Misspecification robustness:** We conducted a new experiment on misspecification robustness in response to Reviewer pkTY’s concerns. Please take a look.
>
>
> - **Necessary domain knowledge makes the framework less generalizable:** We agree that crafting the most salient covariates can be beneficial. In practice, users can start with generic, off-the-shelf metrics (e.g., response length, readability grade) that already capture a large proportion of common biases. For specialized tasks or higher requirements, practitioners may simply augment this base set using their domain knowledge or by consulting subject-matter experts. The core method still applies without modification.
>
> - **Human judgement are treated as gold standard, even if they can also contain biases:** Thank you for raising this. Indeed, like most evaluation pipelines in ML, we have followed the common practice of using human judgments as a proxy for "ground truth", but we fully agree that these judgments can themselves be biased. Nonetheless, our framework is designed to reduce inconsistencies between any chosen benchmark and LLM judgments, whether that benchmark remains individual human ratings or evolves toward multi-annotator consensus or other proxies. This highlights the significance of our work.
>
> - **Comments on citation practices and Table 1:** Thank you for the constructive comments. We will revise accordingly.
>
> -------
> ### **Open sourcing**
>
> Yes. We will publish the codebase and collected data after the review process.
>
> -------
> ### **Did your method reveal any patterns in which LLMs (e.g., size or training regimen) are more misaligned with humans?**
>
> We see that in Table 2, for example, LLaMa-3.1-8b-Instruct has many more detected discrepancies when compared to either fine-tuned models or a notably stronger model (GPT-4o-mini). Moreover, we have run an extra experiment in which we show that LLMs systematically value creativity less than human in less technical queries while the same fact does not hold on technical ones. We explain below.
>
> We conducted an additional experiment by dividing Chatbot Arena queries into two categories. Using GPT-4o-mini as a zero-shot classifier, we split queries into technical (e.g., coding, STEM) and non-technical (e.g., jokes, casual conversation) groups. Approximately half of the queries were labeled as technical. For technical queries, we did not observe any statistically significant differences between human and LLM ratings. However, for non-technical queries, we found notable divergences between humans and LLMs in terms of text length, bold text, and creativity. This suggests that human and LLM judgments are more likely to diverge on subjective, non-technical content. A limitation of this analysis is the reduced sample size in each split, which lowers statistical power.
>
>
> *Chatbot-Arena (Non-technical queries)*
>
> | Feature     | GPT4.1-nano | GPT4.1    | GPT4o-mini | LLaMa-3.1-8B-It | Selene-1-Mini | Prometheus-v2 |
> | ----------- | ----------- | --------- | ---------- | --------------- | ------------- | ------------- |
> | Text Length | **-2.18**** | **-0.55*** | **-1.09****  | -2.45           | -1.25         | -1.67         |
> | Creativity  | **-1.33**** | **-0.33*** | **-0.70****  | -1.64           | -0.86         | -1.07         |
> | Bold Text   | 0.75        | 0.24      | **0.52*** | 1.01            | 0.66          | 0.77          |

---

> > ### Comment · Reviewer_gh7b · 2025-08-01
> > **Answer to Rebuttal by Authors**
> >
> > Thank you for considering my comments and answering my questions. As I was already convinced that this paper is solid and should be accepted for the conference, I will not change my score.

---

### Decision · Program_Chairs · 2025-09-17

**Decision:**

Accept (poster)

**Comment:**

# Summary
LLM-as-a-judge do not always align with human annotators, exhibiting systematic and undesired differences, e.g., biases toward certain writing styles. This paper proposes a statistical model to analyze the discrepancies between LLM judges' outputs (in the form of scores or comparisons) and human evaluations. Based on this model, the authors design a calibration method for LLM judges and conduct hypothesis testing on various covariates used in LLM evaluations. A notable and surprising conclusion drawn from the analysis is that LLM judges favor brevity relative to human raters, which stands in contrast to prior findings.

# Strengths
* The paper is well written and structured.
* It addresses a crucial issue - the biases of LLM judges - which directly impacts the reliability and fairness of LLM-based evaluations
* The paper reports several intriguing findings, most notably that LLM judges tend to prefer shorter responses relative to human raters.
* This challenges previous assumptions and adds valuable insight to the ongoing discourse.

# Weaknesses
* Necessary domain knowledge makes the framework less generalizable
* Human judgement is treated as gold standard, even if they can also contain biases.

# Overall
The paper is well written and there are no major concerns left after the author-reviewer discussion period. A few new experimental results were presented (e.g. results on larger models) and it would be great to include them into the paper.